# MENTOR: Mixture-of-Experts Network with Task-Oriented Perturbation for Visual Reinforcement Learning

Suning Huang [* 1 2]  Zheyu Zhang [* 1 3]  Tianhai Liang [1]  Yihan Xu [1]  Zhehao Kou [1]  Chenhao Lu [1]  Guowei Xu [1]
Zhengrong Xue [1]  Huazhe Xu [1]

## Abstract

Visual deep reinforcement learning (RL) enables robots to acquire skills from visual input for un-structured tasks. However, current algorithms suffer from low sample efficiency, limiting their practical applicability. In this work, we present MEN-TOR, a method that improves both the *architecture* and *optimization* of RL agents. Specifically, MENTOR replaces the standard multi-layer perceptron (MLP) with a mixture-of-experts (MoE) backbone and introduces a task-oriented perturbation mechanism. MENTOR outperforms state-of-the-art methods across three simulation benchmarks and achieves an average of 83% success rate on three challenging real-world robotic manipulation tasks, significantly surpassing the 32% success rate of the strongest existing model-free visual RL algorithm. These results underscore the importance of sample efficiency in advancing visual RL for real-world robotics. Experimental videos are available at mentor-vrl.

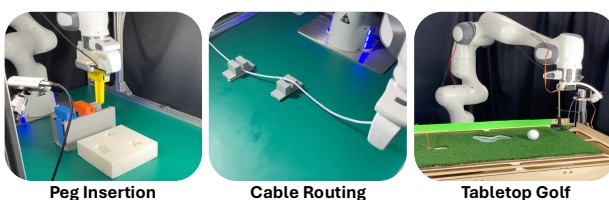

**Peg Insertion**    **Cable Routing**    **Tabletop Golf**

*Figure 1.* **MENTOR validation on real-world tasks.** We design three challenging robotic learning tasks where agents are **trained from scratch in the real world**. MENTOR demonstrates the most efficient and robust policies compared to baseline methods.

---
[*]Equal contribution  [1]Tsinghua University  [2]Stanford University  [3]University of Illinois Urbana-Champaign. Correspondence to: Suning Huang <suning@stanford.edu>, Zheyu Zhang <zheyuz5@illinois.edu>.

*Proceedings of the 42nd International Conference on Machine Learning*, Vancouver, Canada. PMLR 267, 2025. Copyright 2025 by the author(s).

## 1. Introduction

Visual deep reinforcement learning (RL) focuses on agents that perceive their environment through high-dimensional image data, closely aligning with robot control scenarios where vision is the primary modality. Despite substantial progress in this field (Kostrikov et al., 2020; Yarats et al., 2021; Schwarzer et al., 2020; Stooke et al., 2021; Laskin et al., 2020b), these methods still suffer from low sample efficiency. As a result, most visual RL pipelines have to be first trained in the simulator and then deployed to the real world, inevitably leading to the problem of sim-to-real gap (Zhao et al., 2020; Salvato et al., 2021).

To bypass this difficulty, one approach is to train visual RL agents from scratch on physical robots, which is known as real-world RL (Dulac-Arnold et al., 2019; Luo et al., 2024; Zhu et al., 2020). Given the numerous challenges of real-world RL, we argue that the fundamental solution lies not in task-specific tweaks, but in developing substantially more sample-efficient RL algorithms. In this paper, we introduce **MENTOR**: **M**ixture-of-**E**xperts **N**etwork with **T**ask-**O**riented perturbation for visual **R**einforcement learning, which significantly boosts the sample efficiency of visual RL through improvements in both agent network *architecture* and *optimization*.

In terms of architecture, visual RL agents typically use convolutional neural networks (CNNs) for feature extraction from high-dimensional images, followed by multi-layer perceptrons (MLPs) for action output (Yarats et al., 2021; Zheng et al., 2023; Cetin et al., 2022; Xu et al., 2023). However, the learning efficiency of standard MLPs is hindered by intrinsic *gradient conflicts* in challenging robotic tasks (Yu et al., 2020a; Liu et al., 2023; Zhou et al., 2022; Liu et al., 2021), where the gradient directions for optimizing neural parameters across different stages of the task trajectory or between tasks may conflict. In this work, we propose to alleviate gradient conflicts by integrating mixture-of-experts (MoE) architectures (Jacobs et al., 1991; Shazeer et al., 2017; Masoudnia & Ebrahimpour, 2014) as the backbone to the visual RL framework. Intuitively, MoE architectures can alleviate gradient conflicts due to their ability to dynamically allocate gradients to specialized experts for

each input through the sparse routing mechanism (Akbari et al., 2023; Yang et al., 2024).

In terms of optimization, visual RL agents often struggle with local minima due to the unstructured nature of robotic tasks. Recent works have shown that periodically perturbing the agent's weights with random noise can help escape local minima (Nikishin et al., 2022; Sokar et al., 2023; Xu et al., 2023; Ji et al., 2024). However, the choice of perturbation candidates (i.e., the network weights used to perturb the current agent's weights) has not been thoroughly explored. Building on this idea, we propose a task-oriented perturbation mechanism. Instead of sampling from a fixed distribution, we maintain a heuristically shifted distribution based on the top-performing agents from the RL history. The intuition is that the distribution gradually formed by the weights of previous top-performing agents may accumulate task-relevant information, leading to more promising optimization directions than purely random noise.

Empirically, we find MENTOR outperforms current state-of-the-art methods (Xu et al., 2023; Yarats et al., 2021; Cetin et al., 2022; Zheng et al., 2023) across all tested scenarios in *DeepMind Control Suite* (Tassa et al., 2018), *Meta-World* (Yu et al., 2020b), and *Adroit* (Rajeswaran et al., 2017). Furthermore, we present three challenging real-world robotic manipulation tasks, shown in Figure 1: *Peg Insertion* – inserting three kinds of pegs into the corresponding sockets; *Cable Routing* – maneuvering one end of a rope to make it fit into two non-parallel slots; and *Tabletop Golf* – striking a golf ball into the target hole while avoiding getting stuck into the trap. In these experiments, MENTOR demonstrates significantly higher learning efficiency, achieving an average success rate of 83%, compared to 32% for the state-of-the-art counterpart (Xu et al., 2023) within the same training time. This confirms the effectiveness of our approach and underscores the importance of improving sample efficiency for making RL algorithms more practical in robotics applications.

Our key contributions are threefold. First, we introduce the MoE architecture to replace the MLP as the agent backbone in model-free visual RL, improving the agent's learning ability to handle complex robotic environments and reducing gradient conflicts. Second, we propose a task-oriented perturbation mechanism which samples candidates from a heuristically updated distribution, making network perturbation a more efficient and targeted optimization process compared to the random parameter exploration used in previous RL perturbation methods. Third, we achieve state-of-the-art performance in both simulated environments and three challenging real-world tasks, highlighting the sample efficiency and practical value of MENTOR.

## 2. Preliminary

**Mixture-of-Experts (MoE).** Mixture-of-experts (MoE), introduced by Jacobs et al. (1991) and Jordan & Jacobs (1994), is a framework where specialized model components, called experts, handle different tasks or aspects of a task. A sparse MoE layer comprises multiple experts and a router. The router predicts a probability distribution over the experts, activating only the top-$k$ for each input (Shazeer et al., 2017). With $N$ experts, each being a feed-forward network (FFN), the MoE output is:

$$w(i; \mathbf{x}) = \text{softmax}\left(\text{topk}\left(h(\mathbf{x})\right)\right)[i], \qquad (1)$$

$$F^{\text{MoE}}(\mathbf{x}) = \sum_{i=1}^{N} w(i; \mathbf{x}) \, \text{FFN}_i(\mathbf{x}), \qquad (2)$$

where $w(i; \mathbf{x})$ is the gating function determining the weight of the $i$-th expert for input $\mathbf{x}$, $h(\mathbf{x})$ provides logits for expert selection, and $\text{topk}\left(h(\mathbf{x})\right)$ selects the top $k$.

**Visual Reinforcement Learning.** We employ visual reinforcement learning (RL) to train policies for robotic systems, modeled as a Partially Observable Markov Decision Process (POMDP) defined by the tuple $(S, O, A, P, r, \gamma)$. Here, $S$ is the true state space, $O$ represents visual observations, $A$ is the robot's action space, $P : S \times A \to S$ defines the transition dynamics, $r(s, a) : S \times A \to \mathbb{R}$ specifies the reward, and $\gamma \in (0, 1]$ is the discount factor. The goal is to learn an optimal policy $\pi_\theta(a_t \mid o_t)$ that maximizes the expected cumulative reward $E_\pi \left[\sum_{t=0}^{\infty} \gamma^t r(s_t, a_t)\right]$.

**Dormant-Ratio-based Perturbation in RL.** The concept of dormant neurons, introduced by Sokar et al. (2023), refers to neurons that have become nearly inactive. It is formally defined as follows:

**Definition 2.1.** Consider a fully connected layer $l$ with $N^l$ neurons. Let $\text{linear}_i^l(\boldsymbol{x})$ denote the output of neuron $i$ in layer $l$ for an input distribution $\boldsymbol{x} \in \mathcal{I}$. The **score** of neuron $i$ is given by:

$$s_i^l = \frac{\mathbb{E}_{\boldsymbol{x} \in \mathcal{I}} |\text{linear}_i^l(\boldsymbol{x})|}{\frac{1}{N^l} \sum_{k \in l} \mathbb{E}_{\boldsymbol{x} \in \mathcal{I}} |\text{linear}_k^l(\boldsymbol{x})|} . \qquad (3)$$

This neuron is considered $\tau$-dormant if its score $s_i^l \leq \tau$.

**Definition 2.2.** In layer $l$, the total number of $\tau$-dormant neurons is denoted by $D_\tau^l$. The $\tau$-**dormant ratio** of a neural network $\theta$ is defined as:

$$\beta_\tau = \frac{\sum_{l \in \theta} D_\tau^l}{\sum_{l \in \theta} N^l} . \qquad (4)$$

As shown by Xu et al. (2023); Ji et al. (2024), the dormant ratio is a key metric in neural network behavior and enhances

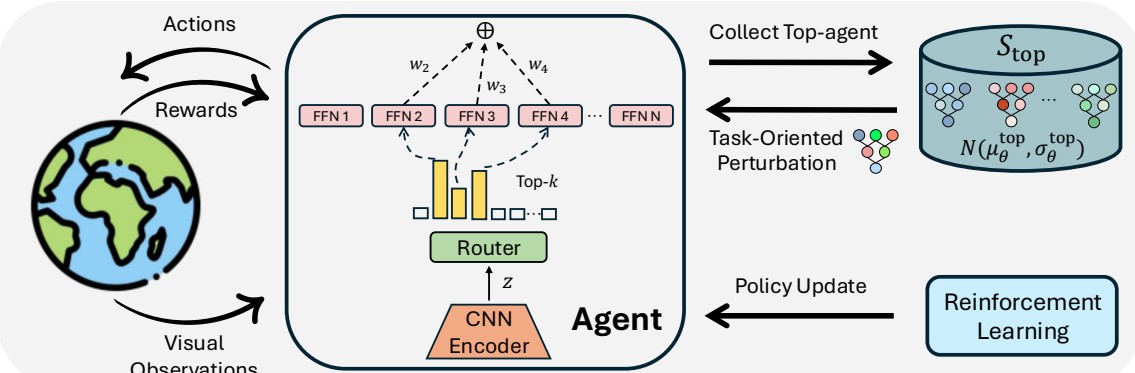

Figure 2. **Overview.** MENTOR uses an MoE backbone with a CNN encoder to process visual inputs. A router selects and weights the relevant experts based on the inputs to generate the final actions. In addition to regular reinforcement learning updates, periodic task-oriented perturbations are applied during training by sampling from top-performing agents to adjust the current agent's weights.

RL efficiency via parameter perturbation. This periodically resets network weights by interpolating between current parameters and random initialization (Ash & Adams, 2020; D'Oro et al., 2022):

$$\theta_k = \alpha\theta_{k-1} + (1-\alpha)\phi, \quad \phi \sim \text{initializer} . \quad (5)$$

Here, $\alpha$ is the perturbation factor, $\theta_{k-1}$ and $\theta_k$ are the weights pre- and post-reset, and $\phi$ denotes randomly initialized weights (e.g., Gaussian noise). The value of $\alpha$ dynamically adjusts as $\alpha = \text{clip}(1-\mu\beta, \alpha_{\min}, \alpha_{\max})$, where $\mu$ is the perturbation rate and $\beta$ the dormant ratio.

## 3. Method

In this section, we introduce MENTOR, which includes two key enhancements to the *architecture* and *optimization* of agents, aimed at improving sample efficiency and overall performance in visual RL tasks. The first enhancement addresses the issue of low sample efficiency caused by gradient conflicts in challenging scenarios, achieved by adopting an MoE structure in place of the traditional MLP as the agent backbone, as detailed in Section 3.1. The second enhancement introduces a task-oriented perturbation mechanism that optimizes the agent's training through targeted perturbations, effectively balancing exploration and exploitation, as outlined in Section 3.2. The framework of our method is illustrated in Figure 2.

### 3.1. Mixture-of-Experts as the Policy Backbone

In challenging robotic tasks, RL agents often assigned $K \geq 2$ different tasks or subgoals, each with a loss $L_i(\theta)$. The goal is to optimize shared weights $\theta \in \mathbb{R}^m$ by minimizing the average loss:

$$\theta^* = \arg\min_{\theta \in \mathbb{R}^m} \left\{ L_0(\theta) \triangleq \frac{1}{K} \sum_{i=1}^{K} L_i(\theta) \right\} .$$

When using shared parameters $\theta$ (e.g., MLP), meaning all parameters must be simultaneously active to function, the optimization process using gradient descent may compromise individual loss optimization. This issue, known as conflicting gradients (Yu et al., 2020a; Liu et al., 2021), hinders the agent's ability to optimize its behavior when facing complex scenarios effectively.

We propose replacing the MLP with an MoE backbone. The MoE, composed of modular experts $\theta_{\text{MoE}} = \{\theta_1, \theta_2, \ldots, \theta_N\}$, which allows the agent to activate different experts via a dynamic routing mechanism flexibly. This enables gradients dynamically route from different tasks to specific experts, reducing gradient conflicts. Each expert is updated using gradients from related tasks, addressing conflicts effectively. As shown in Figure 2, the MoE agent uses a CNN encoder to map visual inputs to latent space $Z$. The router $h$ computes a probability $h(i \mid z)$ over experts for latent $z \in Z$. The top-$k$ experts are selected, and their outputs $a_i$ are combined using softmax weights $w_i$ (Equations 1 and 2). This structure routes inputs to specialized experts, improving multi-task performance.

To illustrate the important role of dynamic modular expert learning for RL agents, we conduct a multi-task experiment (MT5) in Meta-World, training an agent (#Experts = $16, k = 4$) for five opposing tasks: Open (Door-Open, Drawer-Open, Window-Open) and Close (Drawer-Close, Window-Close). Figure 3a shows Open and Close tasks share some experts but also utilize dedicated ones. To quantitatively demonstrate how much the MoE alleviates the gradient conflict issue, we evaluate the cosine similarities (Yu et al., 2020a) for both MLP and MoE agents in Figure 3b. The MLP's gradients show significant conflicts between opposing tasks, while the MoE model demonstrates higher gradient compatibility. As a result, there is a performance gap, with the MLP achieving 100% success in Close tasks

but only 82% in Open tasks, whereas the MoE achieves 100% success in **both** task types.

This structural advantage can also be propagated to challenging single tasks, as the dynamic routing mechanism automatically activates different experts to adjust the agent's behavior throughout the task, alleviating the burden on shared parameters. We illustrate this through training a same-structure MoE agent on a single, highly challenging Assembly task from Meta-World (MW). Figure 4 shows the engagement of the $k = 4$ most active experts during task execution, with Expert 15 serving as the shared module throughout the entire policy execution. The other experts vary and automatically divide the task into four distinct stages: Expert 9 handles gripper control for grasping and releasing; Expert 13 manages arm movement while maneuvering the ring; and Expert 14 oversees the assembly process as the ring approaches its fitting location. More detailed results about how MoE alleviates gradient conflicts in the single task are shown in Appendix F.

---

**Algorithm 1** Task-Oriented Perturbation Mechanism

---

Initialize the set $S_{\text{top}} = \emptyset$, perturb interval $T_p$
**for** each episode $t = 1, 2, \ldots$ **do**
    Execute policy $\pi_{\theta_t}$ and obtain episode reward $R_t$ and dormant ratio $\beta$
    **if** $|S_{\text{top}}| < N$ **then**
        Add $(\theta_t, R_t)$ to $S_{\text{top}}$
    **else**
        **if** $R_t > \min\{R_i \mid (\theta_i, R_i) \in S_{\text{top}}\}$ **then**
            Replace $(\theta_j, R_j)$ with $(\theta_t, R_t)$, where $j = \arg\min R_j$
        **end if**
    **end if**
    **if** (Number of steps since last perturb) $\geq T_p$ **then**
        Compute mean $\mu_\theta^{\text{top}}$ and standard deviation $\sigma_\theta^{\text{top}}$ from $S_{\text{top}}$
        Sample perturbation weight $\phi \sim \Phi_{\text{oriented}} = \mathcal{N}(\mu_\theta^{\text{top}}, \sigma_\theta^{\text{top}})$
        Calculate perturb factor as in Sokar et al. (2023): $\alpha = \text{clip}(1 - \mu\beta, \alpha_{\min}, \alpha_{\max})$
        Update agent weights: $\theta_t = \alpha\theta_t + (1 - \alpha)\phi$
    **end if**
**end for**

---

### 3.2. Task-oriented Perturbation Mechanism

Neural network perturbation is employed to enhance the exploration capabilities in RL. Two key factors influence the effectiveness of this process $\theta_k = \alpha\theta_{k-1} + (1-\alpha)\phi, \phi \sim \Phi$. $\alpha$ is the perturbation factor controlling the mix between current agent and perturbation candidate weights. $\phi$ represents the perturbation candidate sampled from a distribution $\Phi$, which typically is a fixed Gaussian noise $\mathcal{N}(\mu, \sigma)$. Previous works (Xu et al., 2023; Ji et al., 2024) have investigated the use of the dormant ratio to determine $\alpha$, resulting in improved exploration efficiency (see Section 2). However, the selection of perturbation candidates has not been thoroughly examined. In this work, we propose sampling $\phi$ from a heuristically updated distribution $\Phi_{\text{oriented}}$, generated from

past high-performing agents, to provide more task-oriented candidates that better facilitate optimization.

We define $\Phi_{\text{oriented}}$ as a distribution from which high-performing agent weights can be sampled. This distribution is obtained by maintaining a fixed-size set $S_{\text{top}} = \{(\theta, R)\}$, where $(\theta, R)$ represents an agent with weights $\theta$ and episode reward $R$. The distribution is approximated as $\Phi_{\text{oriented}} = \mathcal{N}(\mu_\theta^{\text{top}}, \sigma_\theta^{\text{top}})$, where $\mu_\theta^{\text{top}}$ and $\sigma_\theta^{\text{top}}$ are the mean and standard deviation of weights in $S_{\text{top}}$. As shown in Figure 2, $S_{\text{top}}$ is updated during training: at episode $t$, if an agent with weights $\theta_t$ achieves reward $R_t$ exceeding the lowest in $S_{\text{top}}$, $(\theta_t, R_t)$ replaces the lowest-reward tuple. This ensures $\Phi_{\text{oriented}}$ reflects current high-performing agents, improving perturbation candidates $\phi$ for subsequent iterations. The pseudocode is in Algorithm 1.

For illustration, we conduct experiments on the Hopper Hop task from the DeepMind Control Suite (DMC), comparing task-oriented perturbation approach to leading model-free visual RL baselines (DrM (Xu et al., 2023) and DrQ-v2 (Yarats et al., 2021)). Our approach solely replaces DrM's perturbation mechanism with task-oriented perturbations. Both our method and DrM outperform DrQ-v2 due to dormant-ratio-based perturbation, but our method achieves faster skill acquisition and maintains a lower, smoother dormant ratio throughout training (Figure 5a and 5b). By directly testing perturbation candidates as agents in the task (Figure 5c), we observe that candidates sampled from $\Phi_{\text{oriented}}$ steadily improve throughout training, sometimes even surpassing the performance of the agent they perturb. This demonstrates that $\Phi_{\text{oriented}}$ progressively captures the optimal weight distribution, rather than simply interpolating from past agents, leading to more targeted optimization. In contrast, perturbation candidates from DrM (initialized with Gaussian noise) consistently yield zero reward, indicating the lack of task-relevant information.

## 4. Experiments

In this section, we present a comprehensive empirical evaluation of MENTOR. Section 4.1 showcases its effectiveness on three simulation benchmarks: DeepMind Control Suite (DMC) (Tassa et al., 2018), Meta-World (MW) (Yu et al., 2020b), and Adroit (Rajeswaran et al., 2017), which feature rich visuals and complex dynamics. MENTOR consistently outperforms leading visual RL algorithms. However, a critical limitation in visual RL research is the over-reliance on simulated environments, raising concerns about practical applicability. To bridge this gap, Section 4.2 validates MENTOR on three challenging real-world robotic tasks, highlighting the importance of real-world testing.

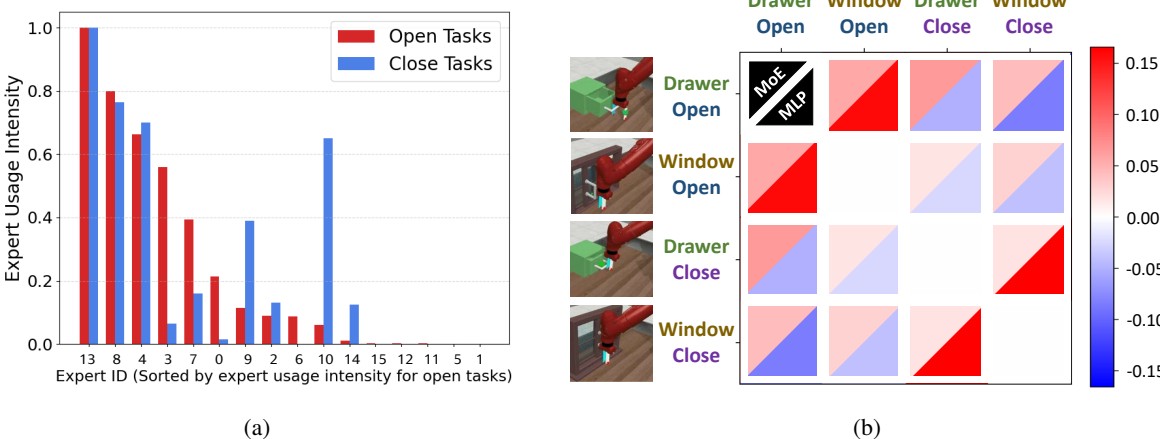

(a)                                                                  (b)

*Figure 3.* **MoE in multi-task scenarios.** Left: Expert usage intensity distribution of the MoE agent in opposing tasks. Right: Gradient conflict among opposing tasks for both MLP and MoE agents. The MLP agent frequently encounters gradient conflicts (indicated by negative cosine similarity) when learning multiple skills, while the MoE agent avoids these conflicts (indicated by positive values). We also provide a comparison of gradient conflicts for MLP and MoE agents in single-task settings, as detailed in Appendix F.

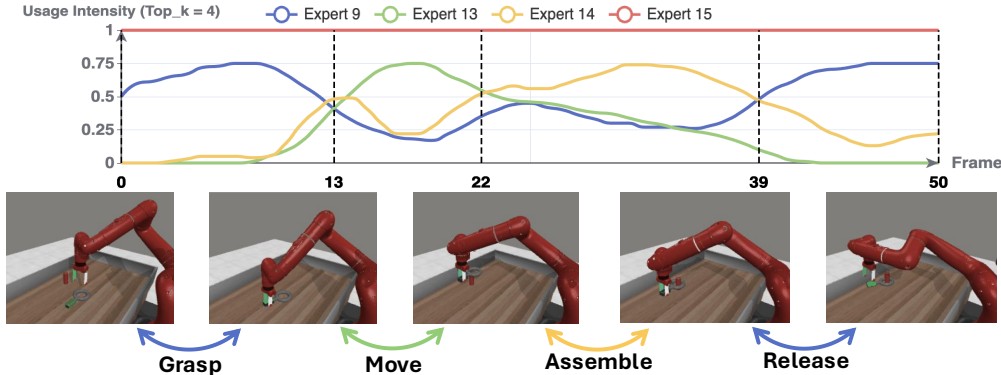

*Figure 4.* **MoE in multi-stage scenarios.** We present the expert usage intensity during the Assembly task in Meta-World. While Expert 15 remains highly active throughout the entire process, other experts are activated with varying intensity over time, automatically dividing the task into four distinct stages.

### 4.1. Simulation Experiments

**Baselines:** We compare MENTOR against four leading model-free visual RL methods: DrM (Xu et al., 2023), ALIX (Cetin et al., 2022), TACO (Zheng et al., 2023), and DrQ-v2 (Yarats et al., 2021). DrM, ALIX, and TACO all use DrQ-v2 as their backbone. DrM periodically perturbs the agent's weights with random noise based on the proportion of dormant neurons in the neural network; ALIX adds regularization to the encoder gradients to mitigate overfitting; and TACO employs contrastive learning to improve latent state and action representations.

**Experimental Settings:** We evaluate MENTOR on a diverse set of tasks across three simulation environments with complex dynamics and even sparse reward. The DMC includes challenging tasks like Dog Stand, Dog Walk, Manipulator Bring Ball, and Acrobot Swingup (Sparse), focusing

on long-horizon continuous locomotion and manipulation challenges. The MW environment provides a suite of robotic tasks including Assembly, Disassemble, Pick Place, Coffee Push (Sparse), Soccer (Sparse), and Hammer (Sparse), which test the agent's manipulation abilities and require sequential reasoning. The Adroit environment includes complex robotic manipulation tasks such as Door and Hammer, which involve controlling dexterous hands to interact with articulated objects. Notably, DMC tasks are evaluated using episode reward, while tasks in MW and Adroit are assessed based on success rate. We conducted experiments with four random seeds on each task, with detailed hyperparameters and training settings provided in Appendix B.

**Results:** Figure 6 presents performance comparisons between MENTOR and the baselines. In the DMC tasks, Dog Stand and Dog Walk feature high action dimensionality with a 38-dimensional action space representing joint controls for

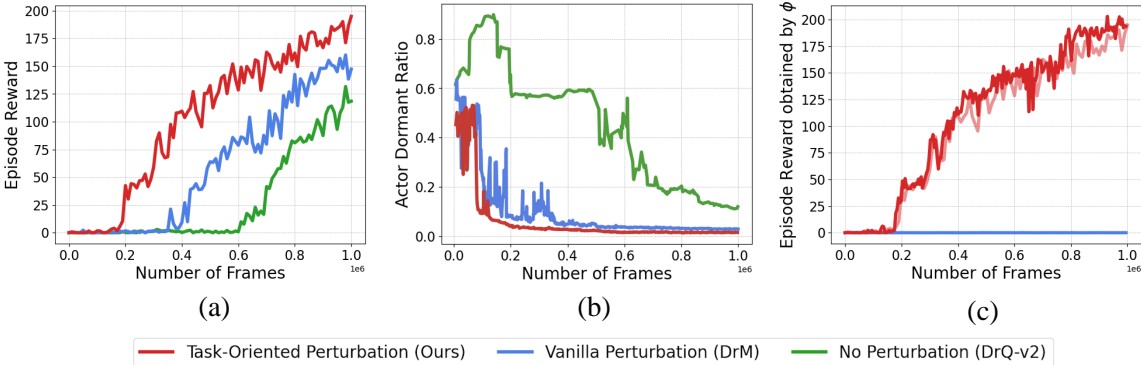

*Figure 5.* **Validation of task-oriented perturbation on Hopper Hop (a MENTOR, DrM, and DrQ-v2 agent trained on the Hopper Hop task during the first 1M frames).** Our method consistently achieves higher episode rewards with a consistently lower dormant ratio. (c) shows the episode reward obtained by perturbation candidate $\phi$ sampled from $\Phi_{\text{oriented}}$ steadily increases and occasionally surpasses that of the corresponding RL agent (replotted as the light red line), whereas in DrM, the reward remains at zero due to the use of randomly generated perturbation parameters.

the dog model. These tasks also have complex kinematics involving intricate joint coordination, muscle dynamics, and collision handling, making them challenging to optimize. Our method outperforms the top baseline, achieving approximately 17% and 10% higher episode rewards, respectively. In the MW tasks, the Hammer (Sparse) task stands out. It requires a robotic arm to hammer a nail into a wall, with highly sparse rewards: success yields significantly larger rewards than merely touching or missing the nail. In fact, the reward for failure is only one-thousandth of the success reward, making the task extremely sparse. However, our task-oriented perturbation effectively captures these sparse rewards, reducing the required training frames by 70% compared to the best baseline. In the Adroit tasks, our method achieves nearly 100% success with significantly less training time, while the most competitive counterpart (DrM) requires more frames, and other baselines fail to match performance even after 6 million frames. A key highlight is the Door task, which involves multiple stages of dexterous hand manipulation—grasping, turning, and opening the door. Leveraging the MoE architecture, our method reduces training time to achieve over 80% success by approximately 23% compared to the best baseline. In summary, MENTOR demonstrates superior efficiency and performance compared to the strongest existing model-free visual RL baselines across all 12 tasks. For the robustness against disturbances of agent trained by MENTOR, please see Appendix I.

**Ablation Study**: We conduct a detailed ablation study to demonstrate the significance contribution of MoE and Task-oriented Perturbation separately in Appendix C.

### 4.2. Real-World Experiments

Our real-world RL experiments evaluate MENTOR on three key challenges: *multi-task learning*, *multi-stage deformable object manipulation*, and *dynamic skill acquisition* with policies **trained from scratch in the real world**.

**Experimental Settings:** We use a Franka Panda arm for execution and RealSense D435 cameras for RGB observations, capturing both global and local views. Rewards are based on the absolute distance between current and desired states. To prevent trajectory overfitting, the end-effector's initial position is randomly sampled within a predefined region at the start of each episode. Tasks are detailed below, with illustrations in Figure 7. Additional details are in Appendix D.

**Peg Insertion:** This task mimics assembly-line scenarios requiring the agent to insert pegs of three shapes (Star, Triangle, Arrow) into corresponding sockets. Simulating contact-rich interactions in these multi-task settings is highly challenging, making real-world evaluation essential.

**Cable Routing:** Manipulating deformable cables presents significant challenges due to the complexities of modeling and simulating their physical dynamics, making this task ideal for direct, model-free visual RL training in real-world environments. The robot must guide a deformable cable sequentially into two parallel slots. Since both slots cannot be filled simultaneously, the agent must perform the task sequentially, requiring long-horizon, multi-stage planning to successfully accomplish the task.

**Tabletop Golf:** In this task, the robot uses a golf club to strike a ball on a grass-like surface, aiming to land it in a target hole. An automated reset system retrieves the ball when it reaches the hole, enters a mock water hazard, or rolls out of bounds, and randomly repositions it. The agent must learn to approach the ball, control the club's striking force and direction to guide the ball toward the hole while avoiding obstacles through real-world interaction.

**Results:** Our policies exhibit strong performance during

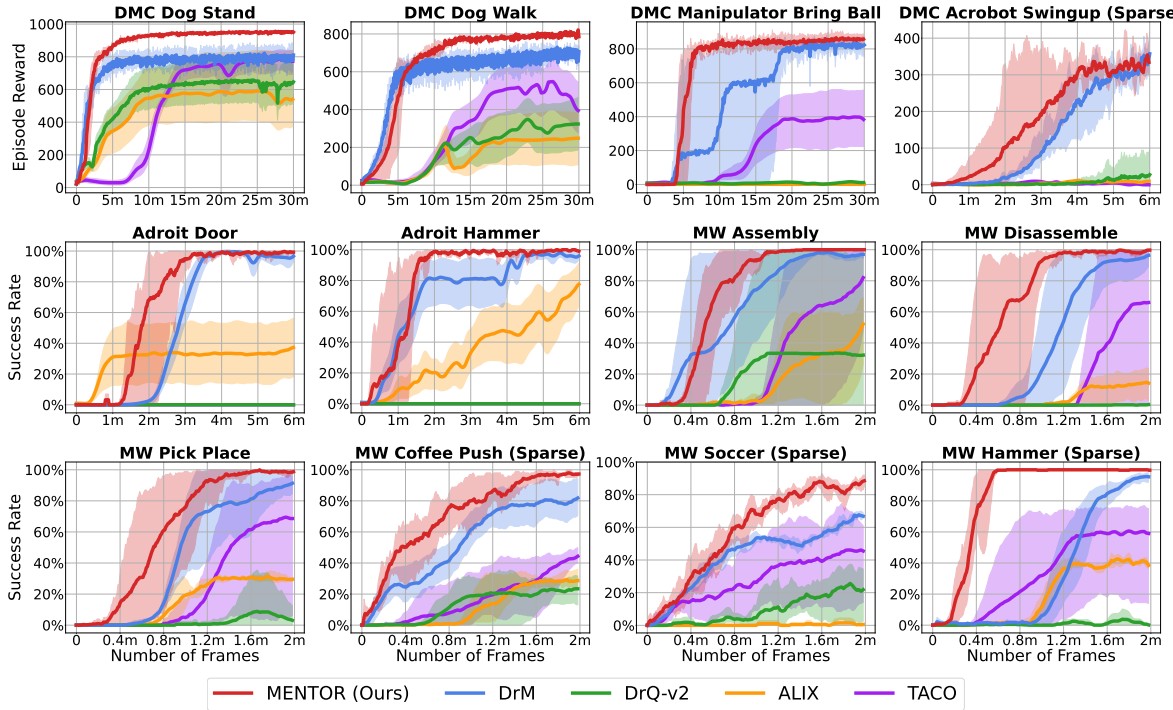

*Figure 6.* **Performance comparisons in simulations.** This figure compares the performance of our method to DrM, DrQ-v2, ALIX, and TACO across 12 tasks with four random seeds in three different benchmarks (DMC, MW, and Adroit). The shaded region indicates standard deviation in DMC and the range of success rates in MW and Adroit.

evaluation, as shown in Figure 7. In Peg Insertion, the agent randomly picks a peg and inserts it from varying initial positions, learning to align the peg with the hole and adjust the angle for accurate insertion. During one execution, as the peg nears the hole, we manually disturb by altering the robot arm's pose significantly. Despite this interference, the agent successfully completes the task relying solely on visual observations. In Cable Routing, the agent learns to prioritize routing it into the farther slot first, then into the closer one. This second step requires careful handling to avoid dislodging the cable from the first slot. During execution, if the cable is randomly removed from the slot, the agent can visually detect this issue and re-route it back into position. In Tabletop Golf, the agent must master two key skills: striking the ball with the correct direction and force, and repositioning the club to follow the ball after the strike. Due to a "water hazard", the ball cannot be struck directly toward the target hole from its starting position. The agent learns to angle its shots to bypass hazards and guide the ball into the hole, even as the ball's rolling on the grass-like surface naturally introduces significant variability.

**Ablation Study:** We conduct a detailed ablation study to demonstrate the effectiveness of MENTOR in improving sample efficiency and performance, as shown in Table 1.

The first two rows reveal that utilizing the pretrained visual encoder (Lin et al., 2024) instead of a CNN trained

from scratch results in an average performance improvement of 9%. However, no significant performance gain is observed in simulation benchmarks with this substitution. This discrepancy may arise from the gap between simulation and real-world environments, where real scenes offer richer textures more aligned with the pretraining domain.

Furthermore, the results confirm the effectiveness of our technical contributions. When the MoE structure is removed from the agent (i.e., replaced with an MLP, as in MENTOR w/o MoE), overall performance drops by nearly 30%. Additionally, further switching the task-oriented perturbation mechanism to basic random perturbation (as in DrM) leads to an additional performance decline of approximately 30%.

## 5. Related work

**Visual Reinforcement Learning.** Visual reinforcement learning (RL), which operates on pixel observations rather than ground-truth state vectors, faces significant challenges in decision-making due to the high-dimensional nature of visual inputs and the difficulty in extracting meaningful features for policy optimization (Ma et al., 2022; Choi et al., 2023; Ma et al., 2022). Despite these challenges, there has been considerable progress in this area. Methods such as Hafner et al. (2019; 2020; 2023); Hansen et al. (2022) improve visual RL by building world models. Other approaches (Yarats et al., 2021; Kostrikov et al., 2020; Laskin

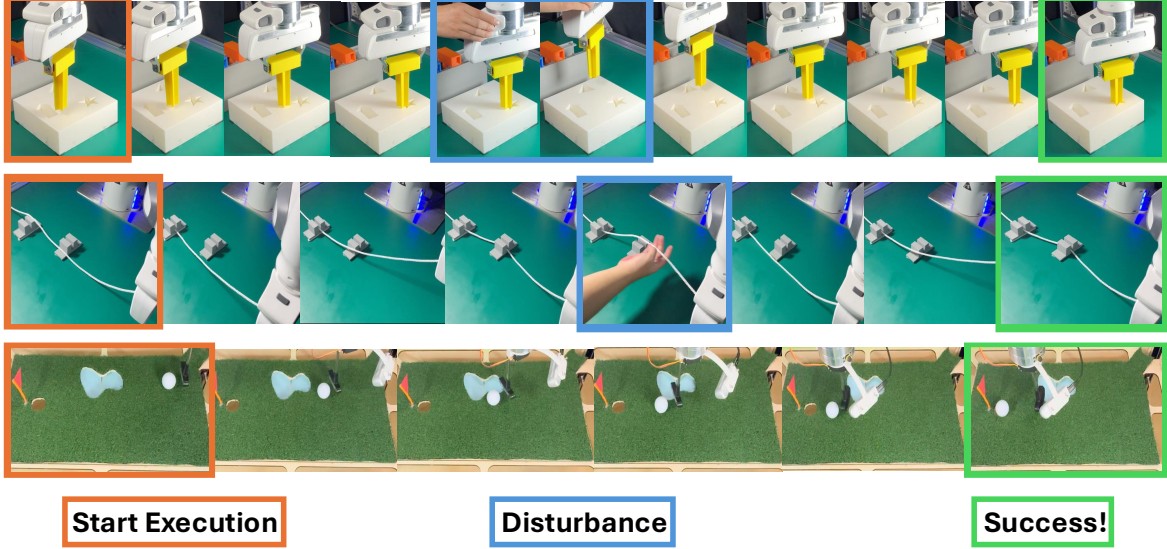

**Start Execution**      **Disturbance**      **Success!**

*Figure 7.* **Real-world experiments (up to down rows: Peg Insertion, Cable Routing, and Tabletop Golf).** This set of images illustrates the execution of the learned visual policy trained using MENTOR. The agent consistently and accurately accomplishes tasks even in the presence of human disturbances.

*Table 1.* **Comparison of success ratios between MENTOR and ablations with equal training times.** Peg Insertion and Cable Routing are trained for 3 hours, and Tabletop Golf for 2 hours. During evaluation, each subtask in Peg Insertion is rolled out 10 times, while Cable Routing and Tabletop Golf are rolled out 20 times.

| Method | Peg Insertion (Subtasks) | | | Cable Routing | Tabletop Golf |
|---|---|---|---|---|---|
| | **Star** | **Triangle** | **Arrow** | | |
| MENTOR w/ pretrained encoder | **1.0** | **1.0** | **1.0** | **0.9** | **0.8** |
| MENTOR | **1.0** | **1.0** | **1.0** | 0.8 | 0.7 |
| MENTOR w/o MoE | **1.0** | 0.7 | 0.6 | 0.45 | 0.55 |
| DrM | 0.5 | 0.2 | 0.1 | 0.2 | 0.5 |

et al., 2020a), use data augmentation to enhance learning robustness from pixel inputs. Contrastive learning, as in Laskin et al. (2020b); Zheng et al. (2023), aids in learning more informative state and action representations. Additionally, Cetin et al. (2022) applies regularization to prevent catastrophic self-overfitting, while DrM (Xu et al., 2023) enhances exploration by periodically perturbing the agent's parameters. Despite recent progress, these methods still suffer from low sample efficiency in complex robotic tasks. In this paper, we propose enhancing the agent's learning capability by replacing the standard MLP backbone with an MoE architecture. This dynamic expert learning mechanism helps mitigate gradient conflicts in complex scenarios.

**Neural Network Perturbation in RL.** Perturbation theory has been explored in machine learning to escape local minima during gradient descent (Jin et al., 2017; Neelakantan et al., 2015). In deep RL, agents often overfit and lose expressiveness during training (Song et al., 2019; Zhang et al., 2018; Schilling, 2021). To address this issue, Sokar

et al. (2023) identified a correlation where improved learning capability is often accompanied by a decline in the dormant neural ratio in agent networks. Building on this insight, Xu et al. (2023); Ji et al. (2024) introduced parameter perturbation mechanisms that softly blend randomly initialized perturbation candidates with the current ones, aiming to reduce the agent's dormant ratio and encourage exploration. However, previous works have not fully explored the choice of perturbation candidates. In this work, we uncover the potential of targeted perturbation for more efficient policy optimization by introducing a simple yet effective task-oriented perturbation mechanism. This mechanism samples perturbation candidates from a time-variant distribution formed by the top-performing agents collected throughout RL history.

## 6. Conclusion

In this paper, we present MENTOR, a state-of-the-art model-free visual RL framework that achieves superior performance in challenging robotic control tasks. MENTOR en-

hances learning efficiency through two key improvements in both agent network *architecture* and *optimization*. MENTOR consistently outperforms the strongest baselines across 12 tasks in three simulation benchmark environments. Furthermore, we extend our evaluation beyond simulations, demonstrating the effectiveness of MENTOR in *real-world* settings on three challenging robotic manipulation tasks. We believe MENTOR is a capable visual RL algorithm with the potential to push the boundaries of RL application in real-world robotic tasks. While MENTOR has proven effective, it has been evaluated on single tasks with a single robot embodiment. Future work could scale our method to handle hundreds of tasks or diverse robot embodiments, enabling broader real-world applications.

## Impact Statement

This paper presents work whose goal is to advance the field of Model-free Visual Reinforcement Learning. There are many potential societal consequences of our work, none which we feel must be specifically highlighted here.

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

# Appendix

## A. Algorithm Details

We illustrate the overview framework of MENTOR in Section 3, where we employ two enhancements in terms of agent *structure* and *optimization*: substituting the MLP backbone with MoE to alleviate gradient conflicts when learning complex tasks, and implementing a task-oriented perturbation mechanism to update the agent's weights in a more targeted direction by sampling from a distribution formed by the top-performing agents in training history. The detailed implementation of task-oriented perturbation is shown in Algorithm 1, and the implementation of using MoE as the policy backbone is described as follows:

Algorithm 2 illustrates how MENTOR employs the MoE architecture as the backbone of its policy network. In addition to the regular training process, using MoE as the policy agent requires adding an additional loss to prevent MoE degradation during training—where a fixed subset of experts is consistently activated. The MoE layer computes the output action while simultaneously calculating an auxiliary loss for load balancing (Lepikhin et al., 2020; Fedus et al., 2022). Specifically, we extract the distribution over experts produced by the router for each input. By averaging these distributions over a large batch, we obtain an overall expert distribution, which we aim to keep uniform across all experts. To achieve this, we introduce an auxiliary loss term—the negative entropy of the overall expert distribution (Chen et al., 2023; Shen et al., 2023). This loss reaches its minimum value of $-\log(N_e)$, where $N_e$ is the number of experts in the MoE, when all experts are equally utilized, thus preventing degradation. This auxiliary loss is added to the actor loss and used to update the actor during the RL process.

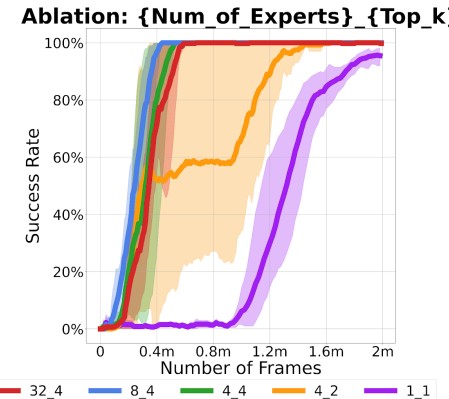

*Figure 8.* **Ablation study on the hyperparameter of MoE.** We evaluated MENTORon MW Hammer (Sparse) wit different MoE settings.

---

**Algorithm 2** Mixture-of-Experts as the Policy Backbone

**Require:** Batch of visual inputs $\{\mathbf{x}_b\}_{b=1}^B$
**Ensure:** Final actions $\{a_b\}_{b=1}^B$, Load balancing loss $\mathcal{L}_{\text{LB}}$
  Initialize experts $\{\text{FFN}_1, \text{FFN}_2, \dots, \text{FFN}_N\}$
  $\{\mathbf{z}_b\}_{b=1}^B \leftarrow \text{Encoder}(\{\mathbf{x}_b\}_{b=1}^B)$
  $\{\mathbf{h}_b\}_{b=1}^B \leftarrow h(\{\mathbf{z}_b\}_{b=1}^B)$
  **for** $b = 1$ to $B$ **do**
    $\mathcal{E}_b \leftarrow \text{topk}(\mathbf{h}_b, k)$
    $w_b(i) \leftarrow \text{softmax}(\mathbf{h}_{b,\mathcal{E}_b})[i], \forall i \in \mathcal{E}_b$
    **for** each $i \in \mathcal{E}_b$ **do**
      $\mathbf{f}_{b,i} \leftarrow \text{FFN}_i(\mathbf{z}_b)$
    **end for**
    $a_b \leftarrow \text{ActionProjector}\left(\sum_{i \in \mathcal{E}_b} w_b(i)\, \mathbf{f}_{b,i}\right)$
  **end for**
  *// Compute load balancing loss*
  $\{\mathbf{pr}_b\}_{b=1}^B \leftarrow \text{softmax}(\{\mathbf{h}_b\}_{b=1}^B)$
  $p(i) \leftarrow \frac{1}{B} \sum_{b=1}^B pr_b(i), \forall i$
  $\mathcal{L}_{\text{LB}} \leftarrow -H(p) = \sum_{i=1}^N p(i) \log(p(i))$

---

## B. Simulation Experimental Settings

The hyperparameters employed in our experiments are detailed in Table 3. In alignment with previous work, we predominantly followed the hyperparameters utilized in DrM (Xu et al., 2023).

For the hyperparameters used in MoE module, Figure 8 shows the ablation study on the number of experts and top_k chosen experts. The results indicate the optimal setting for the Hammer task is MoE has around 8 experts, performance remains consistent across 4, 8, and 32 experts as long as top_k = 4. That is, given a proper top_k, the performance is not sensitive to the number of experts.

## C. Ablation Study on Key Contributions

We conducted additional ablation studies on four diverse tasks: Hopper Hop, Disassemble, Coffee-Push (Sparse), and Hammer (Sparse). These studies aim to decouple the effects of the MoE architecture and the Task-oriented Perturbation (TP) mechanism proposed in our paper.

For the experiments, we evaluate four ablated versions of MENTOR using the same four random seeds as in the original experiments, as shown in Figure 9:

- **MENTOR**: Full model with both MoE and Task-oriented Perturbation.

- **MENTOR_w/o_TP**: Task-oriented Perturbation is replaced with random perturbation.

- **MENTOR_w/o_MoE**: The policy backbone uses an MLP architecture instead of MoE.

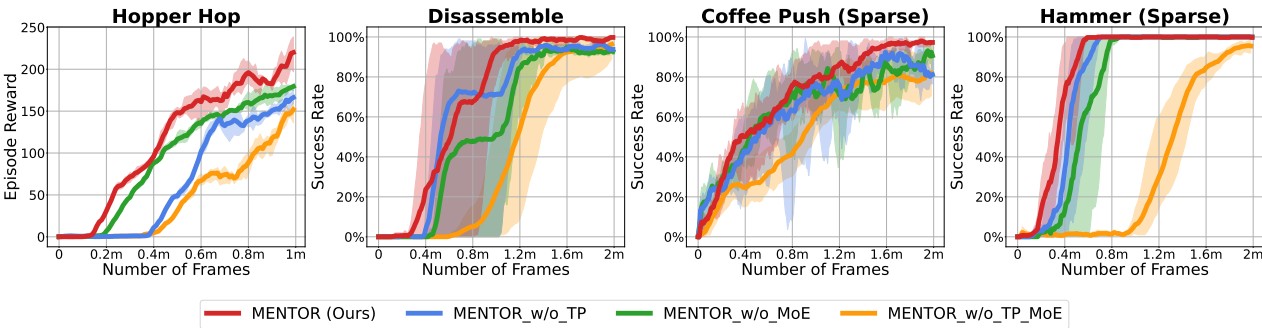

*Figure 9.* **Ablation study on key contributions.** This figure shows the experiment result of four ablated versions of MENTOR using the same four random seeds as in the original experiments, illustrating our Mixture-of-Experts (MoE) and Task-oriented Perturbation (TP) are both significant to improving performance.

*Table 2.* **Sample efficiency comparison across tasks.**

| Sample Efficiency (Training Time) | Hopper Hop | Disassemble | Coffee Push | Hammer |
|:---:|:---:|:---:|:---:|:---:|
| MENTOR (Ours) | **0.6167** | **0.7056** | **0.8066** | **0.7167** |
| MENTOR_w/o_TP | 1 | 0.8505 | 0.9481 | 0.875 |
| MENTOR_w/o_MoE | 0.85 | 1 | 1 | 1 |

- **MENTOR_w/o_TP_MoE**: Neither MoE nor Task-oriented Perturbation is used.

The results, summarized below, demonstrate the individual contributions of each component:

The overall sample efficiency and performance of **MENTOR_w/o_TP** and **MENTOR_w/o_MoE** remain lower than the full **MENTOR** model. This underscores the complementary nature of these two components in enhancing the overall learning efficiency and robustness of **MENTOR**.

Considering the performance, **MENTOR_w/o_MoE** and **MENTOR_w/o_TP** also consistently outperform **MENTOR_w/o_TP_MoE**, indicating that both the MoE architecture and Task-oriented Perturbation independently contribute to improved policy learning.

Considering efficiency, the following part shows a quantitative result based on standard training time.

**Standard Training Time:** Let $T_{\text{MENTOR}}$, $T_{\text{MENTOR\_w/o\_MOE}}$, and $T_{\text{MENTOR\_w/o\_TP}}$ denote the time required for the three different methods to reach the same performance (the final performance of the worst method). The standard training time $T_{\text{standard}}$ is defined as the training time for the worst method to achieve this performance:

$$T_{\text{standard}} = \max(T_{\text{MENTOR}}, T_{\text{MENTOR\_w/o\_MOE}}, T_{\text{MENTOR\_w/o\_TP}})$$

We define normalized sample efficiency as $\frac{T_*}{T_{\text{standard}}}$ (lower is better).

Table 2 shows the normalized sample efficiency of each setting. MENTOR (Ours) achieves an average of **22.6%** and **26.1%** less training time over the 4 tasks compared with MENTOR_w/o_TP and MENTOR_w/o_MoE.

## D. Real-World Experimental Settings

The training and testing videos are available at *mentor-vrl*. The hyperparameters for the real-world experiments are the same as those used in the simulator, as shown in Table 3. We use 16 experts, with the top 4 experts activated.

### D.1. Observation Space

The observation space for all real-world tasks is constructed from information **only** provided by several cameras. Each camera delivers three 84x84x3 images (3-channel RGB, with a resolution of 84x84), which capture frames from the beginning, midpoint, and end of the previous action.

For the Peg Insertion and Tabletop Golf tasks, the observation space is provided by two cameras: a wrist camera and a side camera. As shown in Figure 10, these two cameras in Tabletop Golf offer different perspectives. The wrist camera is attached to the robot arm's wrist, capturing close-up images of the end-effector, while the side camera provides a more global view. As previously mentioned, each camera provides three images, resulting in a total of six 3-channel 84x84 images.

*Table 3.* **Hyper-parameters used in our experiments.**

|  | Parameter | Setting |
|---|---|---|
| Architecture | Features dimension | 100 (Dog) |
|  |  | 50 (Others) |
|  | Hidden dimension | 1024 |
|  | Number of MoE experts | 4 or 16 or 32 |
|  | Activated MoE experts (top-$k$) | 2 or 4 |
|  | MoE experts hidden dimension | 256 |
| Optimization | Optimizer | Adam |
|  | Learning rate | $8 \times 10^{-5}$ (DMC) |
|  |  | $10^{-4}$ (MW & Adroit) |
|  | Learning rate of policy network | 0.5 lr or lr |
|  | Agent update frequency | 2 |
|  | Soft update rate | 0.01 |
|  | MoE load balancing loss weight | 0.002 |
| Perturb | Minimum perturb factor $\alpha_{\min}$ | 0.2 |
|  | Maximum perturb factor $\alpha_{\max}$ | 0.6 (Dog, Coffee Push & Soccer) |
|  |  | 0.9 (Others) |
|  | Perturb rate $\alpha_{\mathrm{rate}}$ | 2 |
|  | Perturb frames | 200000 |
|  | Task-oriented perturb buffer size | 10 |
| Replay Buffer | Replay buffer capacity | $10^6$ |
|  | Action repeat | 2 |
|  | Seed frames | 4000 |
|  | $n$-step returns | 3 |
|  | Mini-batch size | 256 |
|  | Discount $\gamma$ | 0.99 |
| Exploration | Exploration steps | 2000 |
|  | Linear exploration stddev. clip | 0.3 |
|  | Linear exploration stddev. schedule | linear(1.0, 0.1, 2000000) (DMC) |
|  |  | linear(1.0, 0.1, 3000000) (MW & Adroit) |
|  | Awaken exploration temperature $T$ | 0.1 |
|  | Target exploitation parameter $\hat{\lambda}$ | 0.6 |
|  | Exploitation temperature $T'$ | 0.02 |
|  | Exploitation expectile | 0.9 |

In the Cable Routing task, the observation space is constructed using three cameras: a side camera for an overview, and two dedicated cameras for each slot to capture detailed views of the spatial relationship between the slots and the cable. This setup results in a total of nine 3-channel 84x84 images.

### D.2. Action Space

The policy outputs an end-effector delta pose from the current pose tracked by the low-level controller equipped in robot arm. Typically, the end-effector of a robotic arm has six degrees of freedom (DOF); however, in our tasks, the action space is constrained to be fewer. The reason for this restriction in DOF is specific to our setting: in our case, we train model-free visual reinforcement learning algorithms directly in the real-world environment from scratch, without any initial demonstrations and prior knowledge toward the tasks. As a result, the exploration process is highly random, and limiting the degrees of freedom is crucial for safeguarding both the robotic arm and the experimental equipment. For instance, in the Peg Insertion task, the use of rigid 3D-printed materials means allowing the end-effector to attempt insertion at arbitrary angles could easily cause damage. Similarly, in the Cable Routing task, an unrestricted end-effector might collide with the slot, posing a risk to the equipment.

**Peg Insertion:** The end-effector in this task has four degrees

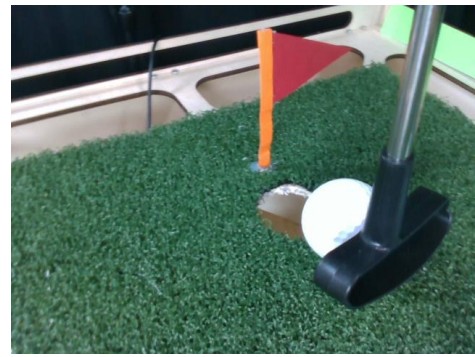
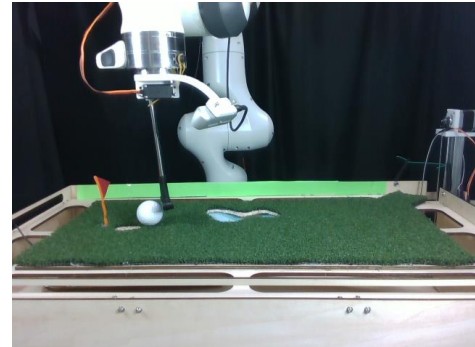

**Wrist Camera View**          **Side Camera View**

*Figure 10.* **Agent's visual observation example in tabletop golf.** MENTOR only uses visual data as policy input. At every step, we capture and stack the frames at the beginning, midpoint, and end of the actuation process. The images captured from cameras are resized to a resolution of 84x84 before being input to the agent.

of freedom: $x$, $y$, $z$, and $r$. Here, $x$ and $y$ represent the planar coordinates, $z$ represents the height, and $r$ denotes the rotation around the $z$-axis. The $x$, $y$, and $z$ dimensions are normalized based on the environment's size, ranging from -1 to 1, while $r$ is normalized over a feasible rotation range of $0.6\pi$.

The action space is a 4-dimensional continuous space $(\Delta x, \Delta y, \Delta z, \Delta r)$, where each action updates the end-effector's state as:

$$(x, y, z, r) \rightarrow \left(x + \frac{\Delta x}{8}, y + \frac{\Delta y}{8}, z + \frac{\Delta z}{10}, r + \frac{\Delta r}{8}\right).$$

**Cable Routing:** In this task, the end-effector is constrained to two degrees of freedom: $x$ and $z$. The $x$-axis controls movement almost perpendicular to the cable, while the $z$-axis controls the height. Both dimensions are normalized based on the environment's size, with values ranging from -1 to 1. Although we restrict the action space to two dimensions, this task remains extremely challenging for the RL agent to master, as it requires inserting cable in both slots sequentially, making it the most time-consuming task among the three, as shown in Figure 14. The difficulty stems largely from the structure and parallel configuration of the two slots: the agent cannot route the cable into both slots simultaneously and must insert one first. However, as shown in Figure 11b, without a hook-like structure to secure the cable in the slot, the cable easily slips out when the agent attempts to route it into the second slot. This task therefore requires highly precise movements, forcing the agent to learn the complex dynamics of soft cables.

The action space is a 2-dimensional continuous space $(\Delta x, \Delta z)$, where each action updates the end-effector's position as:

$$(x, z) \rightarrow \left(x + \frac{\Delta x}{5}, z + \frac{\Delta z}{5}\right).$$

**Tabletop Golf:** The end-effector in this task has three degrees of freedom: $x$, $y$, and $r$. Here, $x$ and $y$ represent the planar coordinates, and $r$ denotes the angle around the normal vector to the $xy$ plane. The $x$ and $y$ dimensions are normalized based on the environment's size, ranging from -1 to 1, while $r$ is normalized over a feasible rotation range of $0.5\pi$.

The action space has four dimensions: three spatial dimensions $(\Delta x, \Delta y, \Delta r)$ and a strike dimension, where the values range from -1 to 1. The end-effector's state is updated as:

$$(x, y, r) \rightarrow \left(x + \frac{\Delta x}{10}, y + \frac{\Delta y}{10}, r + \frac{\Delta r}{8}\right),$$

and if strike $> 0$, the end-effector performs a swing with strength proportional to the value of strike.

### D.3. Reward Design

In this section, we describe the reward functions for the three real-world robotic tasks used in our work: Peg Insertion, Cable Routing, and Tabletop Golf. The basic principle behind these functions is to measure the distance between the current state and the target state. These reward functions are designed to provide continuous feedback—though they can be extremely sparse, as seen in Cable Routing—based on the task's progress, enabling the agent to learn efficient strategies to achieve the goal. Notably, we trained two visual classifiers for the Cable Routing task to determine the relationship between the cables and the slots for reward calculation. Other positional information is obtained through feedback from the robot arm or image processing algorithms. The lower and upper bounds of each dimension in the pose are normalized to -1 and 1, respectively. The coefficients used in the reward functions are listed in Table 4.

**Peg Insertion:** The reward is computed as the negative

absolute difference between the current robot arm pose and the target insertion pose, which varies for each peg.

$$R_{\text{peg}} = \frac{1}{2}\left( \left( \sqrt{2} - \|\mathbf{x}_g - \mathbf{x}_c\| \right) \cdot C_1 + (2 - |\Delta z|) \cdot C_2 \right.$$
$$\left. + \left( \frac{\pi}{2} - |\theta_c - \theta_g| \right) \cdot C_3 - C_4 \right)$$

Where:

- $\mathbf{x}_g$ and $\mathbf{x}_c$: Represent the goal position and the current position of the robot's end-effector in the x-y plane.

- $\|\mathbf{x}_g - \mathbf{x}_c\|$: Euclidean distance between the goal and current positions of the end-effector.

- $\Delta z$: The height difference between the current and target z positions.

- $\theta_c$ and $\theta_g$: Current and goal angles of the end-effector, respectively.

**Cable Routing:** To provide continuous reward feedback, we trained a simple CNN classifier to detect whether the cable is correctly positioned in the slot, awarding full reward when the cable is in the slot and zero when it is far outside. The CNN classifier was trained by labeling images to classify the spatial relationship between the cable and the slot into several categories, with different rewards assigned based on the classification. However, when the cable remains in a particular category without progressing to different stages, the agent receives constant rewards, making it difficult for the agent to learn more refined cable manipulation skills.

$$R_{cable} = r_{\text{slot}_1} + \mathbb{I}(r_{\text{slot}_1} \geq 2) \cdot (r_{\text{slot}_2} + C_5)$$

Where:

- $r_{\text{slot}_1}$: Reward for the first slot, determined by the position of the cable relative to the slot. The possible rewards are:
    - Outside the slot: $r_{\text{slot}_1} = -3$
    - On the side of the slot: $r_{\text{slot}_1} = -1$
    - Above the slot: $r_{\text{slot}_1} = 1$
    - Inside the slot: $r_{\text{slot}_1} = 5$

- $r_{\text{slot}_2}$: Reward for the second slot, with more detailed classifications:
    - Outside the slot: $r_{\text{slot}_2} = -3$
    - On the side of the slot: $r_{\text{slot}_2} = -1$

- Partially above the slot: $r_{\text{slot}_2} = 1$
- Above the slot and at the edge: $r_{\text{slot}_2} = 3$
- Above the slot and close to the middle: $r_{\text{slot}_2} = 5$
- Partially inside the slot: $r_{\text{slot}_2} = 10$
- Fully inside the slot: $r_{\text{slot}_2} = 15$

- $\mathbb{I}(r_{\text{slot}_1} \geq 2)$: Indicator function that activates only if the cable is inserted correctly in the first slot, allowing the agent to receive rewards for the second slot.

**Tabletop Golf:** The reward consists of two components: the negative absolute distance between the robot arm and the ball, and the negative absolute distance between the ball and the target hole. This encourages the agent to learn how to move the robot arm toward the ball and control the striking force and direction to guide the ball toward the hole while avoiding obstacles. Additional rewards include: $R_{\text{golf}} + = C_6$ (if the ball reaches the hole) and $R_{\text{golf}} - = C_7$ (if the ball goes out of bounds). In this experiment, we deploy two cameras at the middle of two adjacent sides of the golf court. The pixel locations of the ball in both cameras are used to roughly estimate its location to calculate the reward function. Despite using an approximate estimation for the reward, MENTOR still quickly learns to follow the ball and strike it with the appropriate angle and force, demonstrating the effectiveness of our proposed method.

$$R_{\text{golf}} = (2 - \|\mathbf{p}_{\text{club}} - \mathbf{p}_{\text{ball}}\|) \cdot C_8 + (2 - \|\mathbf{p}_{\text{ball}} - \mathbf{p}_{\text{hole}}\|) \cdot C_9$$
$$- \mathbb{I}(\text{strike}) + (2 - |\theta_{\text{best}} - \theta_{\text{current}}|) \cdot C_{10}$$
$$- \max(0, \mathbf{p}_{\text{ball}}[y] - \mathbf{p}_{\text{club}}[y] + 0.05) \cdot C_{11}$$

Table 4. **Coefficients used in the reward functions over three real-world robotic tasks.**

| Symbol | Value |
|--------|-------|
| $C_1$ | 16 |
| $C_2$ | 6 |
| $C_3$ | 8 |
| $C_4$ | 17 |
| $C_5$ | 3 |
| $C_6$ | 20 |
| $C_7$ | 5 |
| $C_8$ | 4 |
| $C_9$ | 8 |
| $C_{10}$ | 2 |
| $C_{11}$ | 10 |

Where:

- $\mathbf{p}_{\text{club}}$ and $\mathbf{p}_{\text{ball}}$: Positions of the robot's golf club and the ball, respectively.

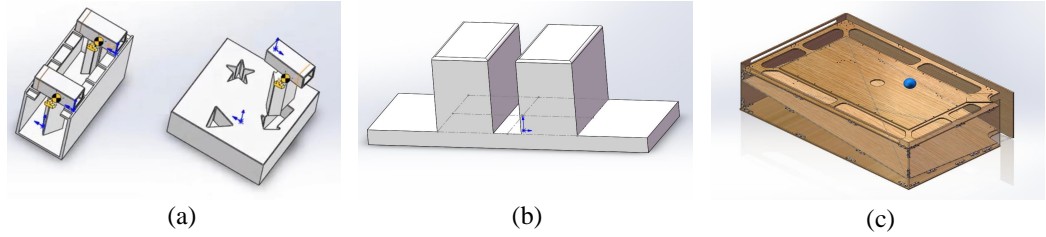

(a)  (b)  (c)

*Figure 11.* **Blueprints of the self-designed mechanisms for the three real-world robotic manipulation tasks (from left to right: Peg Insertion, Cable Routing, and Tabletop Golf).**

*Table 5.* **Comparison of time efficiency in the simulation task (FPS).**

| Task Name | MENTOR | DrM | DrQ-v2 | ALIX | TACO |
|---|---|---|---|---|---|
| Hopper | 37 | 55 | **78** | 49 | 23 |

- $\mathbf{p}_{hole}$: Position of the target hole.

- $\|\mathbf{p}_{club} - \mathbf{p}_{ball}\|$: Distance between the club and the ball.

- $\|\mathbf{p}_{ball} - \mathbf{p}_{hole}\|$: Distance between the ball and the hole.

- $\theta_{best}$ and $\theta_{current}$: Best calculated angle and current angle of the robot's arm for optimal striking.

- $\mathbb{I}(strike)$: Indicator function that penalizes unnecessary strikes.

- $\mathbf{p}_{ball}[y]$ and $\mathbf{p}_{club}[y]$: The y-axis is the long side of the golf course. The ball should be hit from the positive to the negative y-axis, so the club should always be on the positive y-side of the ball.

### D.4. Auto-Reset Mechanisms

One major challenge in real-world RL is the burden of frequent manual resets during training. To address this, we designed auto-reset mechanisms to make the training process more feasible and efficient.

In the Peg Insertion task, the robot arm is set to frequently switch among different pegs to help the agent acquire multi-tasking skills. To facilitate this, we design a shelf to hold spare pegs while the robot arm is handling one. With the fixed position of the shelf, we pre-programmed a peg-switching routine, eliminating the need for manual peg replacement. After switching, the robot arm automatically moves the peg to the workspace and randomizes its initial position for training.

In the Cable Routing task, manual resets are unnecessary, as the robot arm can auto-reset the cable by simply moving back to its initial position with added randomness.

In the Tabletop Golf task, we design an auto-collection mechanism to reset the task. As shown in Figure 11c, the

tabletop golf device has two layers: the top golf court surface and a lower inclined floor. When the ball is hit into the hole or out of bounds, it rolls down to the corner of the lower layer, where a light sensor triggers a motor to return the ball to the court. The variability in the ball's initial velocity during reset introduces randomness to its starting position.

### E. Time Efficiency of MENTOR

We run all simulation and real-world experiments on an Nvidia RTX 3090 GPU and assess the speed of the algorithms compared to baselines. Frames per second (FPS) is used as the evaluation metric for time efficiency.

For simulation, we use the Hopper Hop task to compare time efficiency, as shown in Table 5. While MENTOR demonstrates significant sample efficiency, its time efficiency is relatively lower. This is primarily due to the implementation of a plain MoE version in this work, where input feature vectors are passed to all experts, and only the top-$k$ outputs are weighted and combined to generate the final output. In most tasks, the active expert ratio (i.e., top-$k$/total number of experts) is equal to or below 25%. More efficient implementations of MoE could significantly improve time efficiency, which we leave for future exploration.

We also evaluate time efficiency on three real-world tasks, as shown in Table 6. In real-world applications, the primary bottlenecks in improving time efficiency are data collection efficiency and reset speed. Additionally, the sample efficiency of the RL algorithm plays a crucial role. If the algorithm has low sample efficiency, it may take many poor actions over a long training period, leading to frequent auto-resets and ultimately lowering the overall FPS.

As a result, MENTOR and DrM achieve similar levels of

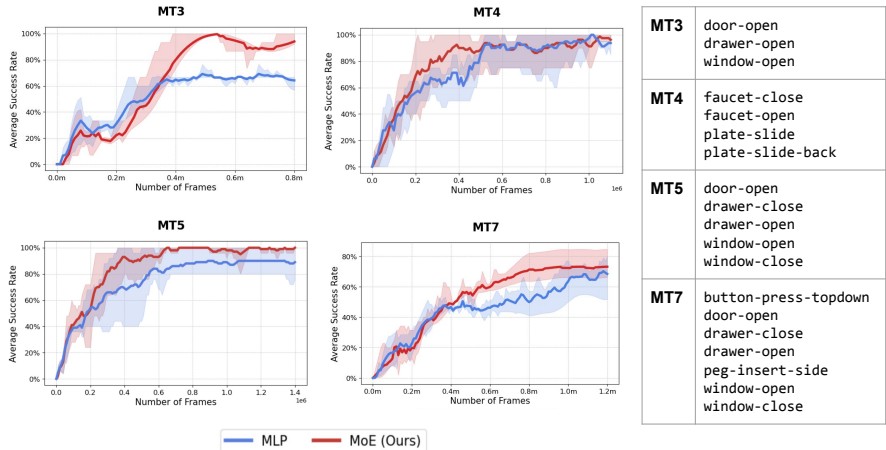

Figure 12. **Multi-task performance of MoE on Meta-World Simulator.** Evaluation accuracy during training for 3, 4, 5, and 7-task combinations, highlighting MoE's advantage over MLP in multitask settings.

Table 6. **Comparison of time efficiency in real-world tasks (FPS).**

| Task Name | MENTOR | DrM |
|---|---|---|
| Peg Insertion | **0.46** | 0.40 |
| Cable Routing | **0.67** | 0.62 |
| Tabletop Golf | **0.52** | 0.47 |

efficiency. However, due to its superior learning capability, MENTOR quickly acquires skills and transitions out of the initial frequent-reset phase faster than DrM, leading to slightly better overall time efficiency during training.

We further extend the training process of the DrM baseline to reach the same performance level as MENTOR in three real-world experiments with the training time comparison shown in Figure 14, which demonstrates an average 37% improvement in time efficiency for our method. These findings underscore the importance of each component in achieving superior results.

# F. Mixture-of-Experts Alleviation Gradient Conflicts in Single Task

In Meta-World, manipulation tasks are associated with compound reward functions that typically include components such as reaching, grasping, and placing. Conflicts between these objectives can arise, creating a burden for shared parameters.

To validate this, we analyze the gradient cosine similarities for the Assembly task. The Assembly task, as shown in Figure 4 can naturally be divided into four stages: Grasp, Move, Assemble, and Release.

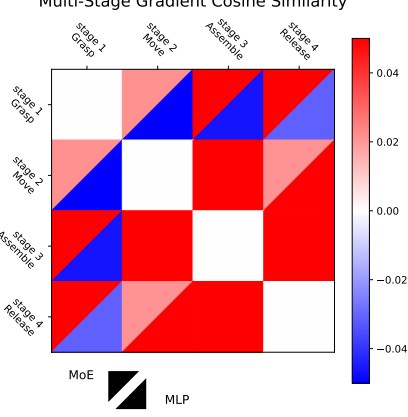

Figure 13. **Cosine similarity of multistage in a single task.** This figure shows the cosine similarities of gradients on the corresponding four stages (Grasp, Move, Assemble, and Release) for both MLP and MoE agents.

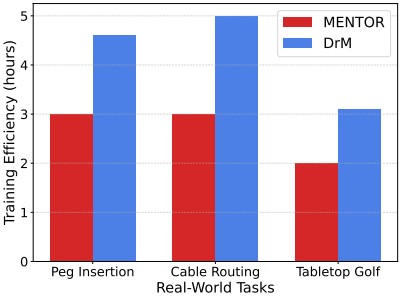

Figure 14. **Time efficiency comparison.** This figure compares the training time required for DrM to reach the performance level of MENTOR, as shown in Table 1.

To illustrate how Mixture-of-Experts alleviates gradient conflicts in a single task, we evaluate the cosine similarities of

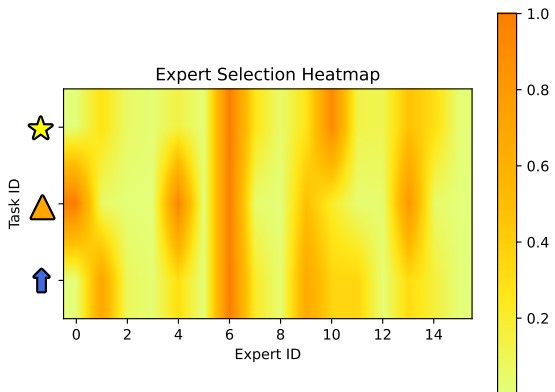

*Figure 15.* **Expert utilization on Peg Insertion task.** This figure shows the usage intensity of the 16 experts in MENTOR during the Peg Insertion task for three different plug shapes.

gradients on the corresponding four stages for both MLP and MoE agents, as shown in Figure 13. The result show that the MLP agent experiences gradient conflicts between grasping and the other stages. This can occur because the procedure of reaching to grasp objects could increase the distance between the robot and the target pillar, leading to competing optimization signals. In contrast, the MoE agent mitigates these conflicts, achieving consistently positive gradient cosine similarities across all stage pairs. This validates the ability of the MoE architecture to alleviate the burden of shared parameters and facilitate more efficient optimization, even in single-task scenarios.

## G. MENTOR in Simulator Multi-Tasking Process

To further demonstrate the multi-task capabilities of MoE, we conducted additional multitask learning experiments on Meta-World. In these experiments, we used four task combinations, consisting of 3, 4, 5, and 7 tasks, respectively. Figure 12 shows the evaluation accuracy during training and the detailed composition of the multi-tasks. In this experiment, neither MENTOR nor DrM used perturbation, with the only difference being the use of MoE versus MLP, indicating the effectiveness of MoE in multi-tasking process.

## H. MENTOR in Real-World Multi-Tasking Process

Figure 15 shows the utilization of experts in the Peg Insertion task for various plug shapes. Each shape is handled by some specialized experts, which aids in multi-task learning. This specialization helps mitigate gradient conflict by directing gradients from different tasks to specific experts, improving learning efficiency, as discussed in the main text.

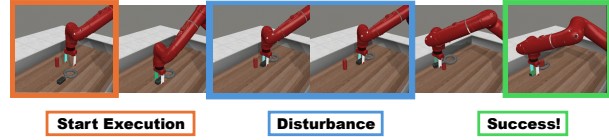

*Figure 16.* **Random disturbances in simulation.** This figure shows the execution of the learned agent using MENTOR. The agent consistently accomplishes Assembly task even with the disturbance.

## I. Random Disturbances in Simulation

To demonstrate the generalization capabilities of the agents trained by MENTOR, we have introduced random disturbances in the real-world experiments presented in Section 4.2. Additionally, we make evaluation of Meta-World Assembly task with random disturbance. In detail, the training phase remains unchanged, but during evaluation, we introduce a random disturbance: after the robot grasps the ring and moves toward the fitting area, the fitting pillar randomly changes its location (Disturbance). This forces the robot agent to adjust its trajectory to the new target position. Figure 16 shows the agent consistently accomplishes Assembly task even with the disturbance, showing the policies learned by MENTOR exhibit strong robustness against disturbances.

