# OpenReview forum: "MENTOR: Mixture-of-Experts Network with Task-Oriented Perturbation for Visual Reinforcement Learning"
_ICML.cc/2025/Conference — ICML 2025 poster_

### Official Review · Reviewer_akYq · 2025-03-07

**Overall Recommendation:** 3

**Summary:**

This paper proposes to improve the sampling efficiency in RL learning by mixture-of-experts (MoE) network design and dormant-based parameter perturbation. The MoE design aims to decrease gradient conflicts and perturbation from top-k agent's parameter helps to accelerate the learning. Experiments in simulator and real world proves the efficiency and effectiveness of the proposed method.

**Claims And Evidence:**

The main claim of this paper is that the introduced method improves the sampling efficiency in RL, which can be validated in two aspects. Firstly, the expert usage intensity distribution shows that different experts are in charge of different skills, which alleviates gradient conflicts. Besides, the parameter perturbation leads to faster learning in experiments and ablations. Therefore, the main claim is supported.

**Essential References Not Discussed:**

No.

**Experimental Designs Or Analyses:**

While the overall evaluation is valid with diverse tasks and environments, I have two concerns.

Firstly, it seems that multi-tasking is the advantage of the proposed method. However, most of the evaluation is on single task. Why not evaluating on multi-task learning (e.g., leaning on all DMC tasks with one model). Besides, the cut-off frames of different task varies a lot (e.g., 2m and 30m). While the complexity of different tasks varies, I believe it's better to show the results after enough frames, especially when considering that the baselines are not converging in some tasks as shown in the figures.

**Methods And Evaluation Criteria:**

Yes. The MoE design and perturbation aims to solve the sampling inefficiency and sparsity in reward. The evaluation uses episode rewards or successful rate as metric.

**Other Comments Or Suggestions:**

1. Evaluation on more tasks and combination of tasks (as said in previous section) could further prove the effectiveness of the method.

2. Please explain more on the two designs as in Weaknesses.

**Other Strengths And Weaknesses:**

### Strengths

1. The motivation of improving sampling efficiency in robot RL makes sense.

2. The design of MoE to distribute learning burden and decrease gradient conflicts makes sense. The parameter perturbation from top agents empirically accelerate the learning process.

2. The writing is clear and easy to follow.

### Weaknesses

1. While the design of two main contributions makes sense separately, I feel that they are contradicting each other. The assumption of MoE is that different agents will master different skills to avoid gradient conflicts, thus improving data efficiency. However, the parameter perturbation uses the mean value of parameters of top k agents. This design implies two messages. First, each agent can independently perform the task, which is not what the usage intensity distribution in single task shows. Furthermore, It means that the agents are somehow similar to each other, which contradict to the benefit of using MoE. I think the two designs are based on contradicting assumptions.

2. I am also concerned about the generalization of MoE model design. The number of experts could be highly dependent on the number and complexity of tasks, which limits the application of this approach on more diverse tasks.

**Questions For Authors:**

Please see the Weaknesses and Evaluation designs. I am willing to raise my rating if the concerns are solved.

**Relation To Broader Scientific Literature:**

It's related to RL in robot manipulation and locomotion. Specifically, robot RL when reward is sparse and exploration space is huge.

**Theoretical Claims:**

Do not apply.

---

> ### Author Rebuttal · Authors · 2025-04-01
>
> > Q1: Evaluation on Multi-task Learning:
>
> We appreciate the reviewer’s suggestion regarding multi-task evaluation. We have conducted additional experiments where our method is trained jointly on Meta-World tasks using a single policy. The performance is compared against MLP baseline under the same multi-task setup. Experimental details and results are provided on the [rebuttal Section A](https://mentor-vrl.github.io/).
>
> > Q2: Cut-off Frames of different tasks:
>
> Thank you for your comment. Our frame settings follow prior works such as DrQv2 [14], TACO [15], and DrM [16], and are chosen to reflect task difficulty while maintaining consistency in comparison. We agree that cut-off frames should account for whether baselines have converged. As noted, our main comparisons are against DrM [16], which consistently converges by the reported cut-off in most tasks.
>
> > Q3: “While the design of two main contributions makes sense separately, I feel that they are contradicting each other.”:
>
> Thanks for pointing out your confusion. We will explain why these two main contributions do not contradict each other.
>
> **For MoE:** we would like to firstly clarify the different concepts of *agent* and *expert*. In our context, *agent* refers to the policy agent, whose backbone is a MoE, and MoE consists of several experts. So the assumption of MoE in your original comment — “different agents will master different skills to avoid gradient conflicts” — should be rephrased as “different experts will master different skills to avoid gradient conflicts”.
>
> **For Task-oriented Perturbation Mechanism:** During RL training, we maintain a fixed-size set $S_{\text{top}} = \{(\theta, R)\}$, which stores the weights $\theta$ and corresponding episode rewards $R$ of the top-performing agents seen so far. At each episode $t$, if the current agent’s reward $R_t$ exceeds the lowest reward in $S_{\text{top}}$, we replace the corresponding tuple with $(\theta_t, R_t)$.
>
> Suppose now we have $N$ history top-performing agent weights (each has $M$ experts) in $S_{\text{top}}$ and we want to execute task-oriented perturbation, the goal distribution to perturbing $\mathrm{expert}_{i}$ is $\mathcal{N}(\mu_i, \sigma_i)$, where $\mu_i$ and $\sigma_i$ are the mean and std of
>
> {exper$\text t_i^{\theta_k}$ $\mid \theta_k \in S_{\text{top}}$},
>
> in these $|S_{\mathrm{top}}|$ agents, and has nothing to do with other experts. So the perturbation process will not make experts similar but will further diversify them from each other.
>
> We hope this is sufficient to clarify that these two designs are not contradictory. More details could be found in the original submission **Section 3.2** and feel free to ask if you have more questions!
>
> > Q4: Generalization and Robustness of MoE Design:
>
> We understand the reviewer’s concern regarding the robustness of the MoE design to hyperparameter choices. To this end, we conducted experiments on the Hammer (Sparse) task, as shown in the [rebuttal Section B](https://mentor-vrl.github.io/) (due to the time limitations, we only report the ablation study on this task), varying the number of experts and top_k. Results show that while the optimal setting is MoE has 8 experts, performance remains consistent across 4, 8, and 32 experts as long as top_k = 4. This suggests the model is not overly sensitive to the number of experts.
>
> ----
> [14] Yarats, D.,et.al. Mastering visual continuous control: Improved data-augmented reinforcement learning. arXiv preprint arXiv:2107.09645.
>
> [15] Zheng, R., Wang, X., Sun, Y., Ma, S., Zhao, J., Xu, H., ... & Huang, F. (2023). TACO: Temporal Latent Action-Driven Contrastive Loss for Visual Reinforcement Learning. Advances in Neural Information Processing Systems, 36, 48203-48225.
>
> [16] Xu, G.,et.al. Drm: Mastering visual reinforcement learning through dormant ratio minimization. arXiv preprint arXiv:2310.19668.

---

> > ### Comment · Reviewer_akYq · 2025-04-03
> >
> > Thanks the authors for their detailed rebuttal. It addresses most of my concerns. Now the method makes more sense to me. I have raised my score to 3.

---

### Official Review · Reviewer_KF8H · 2025-03-12

**Overall Recommendation:** 3

**Summary:**

This paper aims to improve the performance of reinforcement learning agents in robotic tasks. Addressing the issues of gradient conflicts in standard MLPs for robotic tasks and the tendency of visual RL agents to get stuck in local minima, two key improvements are proposed. First, the Mixture-of-Experts (MoE) architecture is used to replace MLP as the policy backbone, reducing gradient conflicts through a dynamic routing mechanism. Second, a task-oriented perturbation mechanism is designed to sample perturbation candidates from a heuristically updated distribution based on past high-performing agents. Experimental results show that the method based on MoE and the new perturbation mechanism outperforms baseline models on three simulation benchmarks and three challenging real-world robotic manipulation tasks.

**Claims And Evidence:**

Yes.

**Essential References Not Discussed:**

Both papers, MENTOR and [1], revolve around the related theme of the multi-expert mixture architecture in reinforcement learning.

[1] Ren, Jie, et al. "Probabilistic mixture-of-experts for efficient deep reinforcement learning." arXiv preprint arXiv:2104.09122 (2021).

**Experimental Designs Or Analyses:**

Yes.

- Extensive experiments were conducted across multiple different tasks and environments, including three simulation benchmarks and three challenging real-world robotic manipulation tasks. Especially in the case of real-world robotics, it encompasses rich aspects such as multi-task learning, multi-stage deformable object manipulation, and dynamic skill acquisition.

- Although ablation experiments were conducted to verify the effectiveness of each component, for some complex components (such as the dynamic routing mechanism in the MoE architecture), the impact of internal parameter changes on the overall performance may not have been further analyzed in depth. For example, the performance differences of different values of (k) (the number of selected experts) in different tasks were not discussed in detail. Additionally, there were no relevant ablation experiments regarding the perturbation factor. What are the bases for determining $\alpha_{min}$ and $\alpha_{max}$?

- Moreover, I am also curious about whether the MoE architecture and the task-oriented perturbation mechanism increase the computational complexity of the algorithm.

**Methods And Evaluation Criteria:**

Yes.

**Other Comments Or Suggestions:**

No.

**Other Strengths And Weaknesses:**

### Strengths
- **Architectural innovation**: Introduces the MoE architecture into model-free visual RL, replacing MLP as the agent backbone. This design solves the gradient conflict problem in robotic tasks via dynamic routing, offering new ways to boost the agent's learning ability in complex environments.
- **Perturbation mechanism innovation**: Proposes a task-oriented perturbation mechanism. Sampling from a heuristically updated distribution (based on past high-performing agents) instead of a fixed one makes perturbation more task-relevant and enhances the optimization and exploration in RL.

### Weakness
- Despite demonstrating MENTOR's effectiveness through experiments, there is insufficient in-depth theoretical analysis and mathematical proof on how the MoE architecture and task-oriented perturbation mechanism reduce gradient conflicts and improve optimization efficiency across different tasks. Relying only on experimental results may not help readers fully grasp the underlying principles.

**Questions For Authors:**

No.

**Relation To Broader Scientific Literature:**

- Previous works (Yu et al., 2020a; Liu et al., 2021) have identified the problem of gradient conflicts in robotic tasks when using shared-parameter architectures like MLPs. In complex robotic scenarios where an agent is assigned multiple tasks or sub-goals, the gradients for optimizing neural parameters across different task stages or between tasks can conflict, hindering the agent's learning ability.

- Previous works (Xu et al., 2023; Ji et al., 2024) have explored using the dormant ratio to determine the perturbation factor $\alpha$, which improved exploration efficiency. However, these works mainly focused on the perturbation factor and did not thoroughly examine the selection of perturbation candidates.

**Theoretical Claims:**

No.

---

> ### Author Rebuttal · Authors · 2025-04-01
>
> > Q1: “The impact of internal parameter changes on the overall performance may not have been further analyzed in depth… Additionally, there were no relevant ablation experiments regarding the perturbation factor.”:
>
> We appreciate the reviewer’s attention to this point. Due to time limitations, we only report the ablation study for the number of experts and top_k in the Hammer (Sparse) task, as shown in the [rebuttal Section B](https://mentor-vrl.github.io/). The results indicate that in the Hammer (Sparse) task, the optimal choice for the number of experts is 8 and for top_k is 4. When top_k is 4, there are no significant performance differences when the number of experts is set to 4, 8, or 32, which suggests that 4 experts are enough to learn the skill in this task. The ablation on top_k further validates our hypothesis, as reducing top_k (to 2) results in a worse learning curve. If the number of experts is set to 1 and top_k is also 1, the MoE will downgrade to a standard MLP, resulting in the worst performance among all configurations.
> Regarding the perturbation hyperparameters, we follow DrM [11] by adopting the same perturbation factor and α values from the open-source github repo.
>
> > Q2: “Whether the MoE architecture and the task-oriented perturbation mechanism increase the computational complexity of the algorithm.”:
>
> We agree that computational efficiency is an important concern. Importantly, both MoE and the perturbation mechanism do not increase the theoretical computational complexity of the algorithm. Empirically, as noted in the original submission **Appendix E**: Time Efficiency of MENTOR, while MoE may introduce higher latency compared to standard MLPs, this is primarily due to hardware-level optimizations (e.g., memory access patterns), rather than additional computation. Recent developments such as MoE-Infinity [12], which demonstrate the potential to reduce MoE latency to negligible levels.
>
> > Q3: “Insufficient in-depth theoretical analysis and mathematical proof”:
>
> We thank the reviewer for the insightful comment. We acknowledge the importance of a thorough theoretical analysis of how the MoE architecture and task-oriented perturbation help reduce gradient conflicts; however, this is beyond the scope of the present work. Concurrent works also investigate similar patterns through an empirical manner, for example, STGC [13] also applied dense experiments to show the use of MoE to mitigate gradient conflicts for NLP tasks.
>
> ----
> [11] Xu, G.,et.al. Drm: Mastering visual reinforcement learning through dormant ratio minimization. arXiv preprint arXiv:2310.19668.
>
> [12] Fu, Y.,et.al. MoE-CAP: Cost-Accuracy-Performance Benchmarking for Mixture-of-Experts Systems. arXiv preprint arXiv:2412.07067.
>
> [13] Yang, L.,et.al. Solving token gradient conflict in mixture-of-experts for large vision-language model. arXiv preprint arXiv:2406.19905.

---

### Official Review · Reviewer_xNhQ · 2025-03-16

**Overall Recommendation:** 4

**Summary:**

This paper proposes MENTOR (Mixture-of-Experts Network with Task-Oriented Perturbation) to improve sample efficiency and performance in visual reinforcement learning (RL). MENTOR replaces the policy’s backbone with a mixture-of-experts (MoE) architecture. This MoE design aims to mitigate gradient conflicts in challenging multi-stage or multi-task settings by allocating different parts of the network (“experts”) to specialized subtasks or states. Additionally, the authors introduce a task-oriented perturbation mechanism that periodically “resets” or “perturbs” the policy parameters in a guided fashion—specifically by sampling from a distribution formed by top-performing past agents instead of adding purely random noise. This approach is shown to offer more directed exploration than baseline methods. Empirically, the paper demonstrates strong gains on three simulation benchmarks (DMC, Meta-World, Adroit) and three real-world robotic tasks (peg insertion, cable routing, and tabletop golf), claiming both superior sample efficiency and higher final performance than prior state-of-the-art methods.

**Claims And Evidence:**

The main claims in this paper are (1) adopting a mixture-of-experts backbone for the policy helps resolve gradient conflicts that arise in complex RL tasks, and (2) task-oriented perturbation—sampling perturbation candidates from a distribution of well-performing agents—improves exploration and stability over purely random perturbation. Both are well supported by clear and convincing evidence, including didactic examples (e.g., Figure 3, 4, 5), as well as comprehensive experiments in both simulation and real world tasks.

**Essential References Not Discussed:**

All essential related works are discussed to the best of my knowledge.

**Experimental Designs Or Analyses:**

Yes. The simulated tasks (DMC, Meta-World, Adroit) are each tested under consistent hyperparameters, showing that MENTOR consistently outperforms baselines. The paper uses multiple seeds (usually four) to report average performance. For real-world tasks, the authors design three testbeds that stress multi-task learning (peg insertion with multiple shapes), sequential multi-stage manipulation (cable routing), and dynamic hitting (tabletop golf). The hardware aspects are well thought out (auto-reset mechanisms, camera angles, etc.) so that training remains feasible without an excessive manual setup.

**Methods And Evaluation Criteria:**

Yes, the benchmarks such as MetaWorld, DMC, are standard in the visual RL literature. Additionally, real-world results are demonstrated, which are quite impressive.

**Other Comments Or Suggestions:**

N/A.

**Other Strengths And Weaknesses:**

Weaknesses:
1. I feel that some more recent benchmarks that emphasize large-scale multi-task, lifelong learning, such as LIBERO, may be better suited to demonstrate the superiority of the proposed approach. The tested benchmarks are a bit saturated.
2. The use of MoE architecture may not be specific to visual RL; it would be good to show why visual RL particularly benefits from this approach. Would state-based RL also benefit?
3. The additional computational burden of the MoE layers is not discussed in details; while MoE can provide sample efficiency, but how does it compare to MLP in terms of wall-clock time as well as compute?

**Questions For Authors:**

My suggestions and questions are listed above.

**Relation To Broader Scientific Literature:**

The authors build directly on the use of data augmentation in visual RL (e.g., DrQ-v2) and on dormant neuron perturbation in DrM. Their main architectural novelty is bringing the mixture-of-experts concept—long used in large-scale language modeling or multi-task learning—to the standard visual RL pipeline.
The results also connect to prior work on multi-task learning where gradient conflicts arise (like conflict-averse gradient descent, gradient surgery, etc.). MENTOR is conceptually aligned with that tradition, providing an RL-specific approach using MoE. Overall, the paper extends known methods (DrQ-v2, DrM, MoE, etc.) in a novel combination well-suited for visual RL.

**Theoretical Claims:**

This paper does not provide any proofs or theoretical claims.

---

> ### Author Rebuttal · Authors · 2025-04-01
>
> > Q1: Regarding the use of more recent benchmarks such as LIBERO that emphasize large-scale multi-task and lifelong learning:
>
> We appreciate the reviewer’s suggestion of more recent benchmarks, like life-long learning tasks. We consider this a valuable avenue for future work. However, we note that this is beyond the current scope of our work, which focuses on evaluating our approach under widely used and established benchmarks and does not consider learning throughout lifetime. Recent works such as Streaming RL [5] and Ace [6] also primarily evaluate on the same or similar settings. Incorporating benchmarks like LIBERO remains an exciting extension for future research direction.
>
> > Q2: Regarding the specificity of the MoE architecture to visual RL:
>
> We agree with the reviewer that the MoE architecture is versatile and not tied to visual RL. In this work, we focus on visual RL because of its growing attention in research ([6][7][8]) and importance in real-world applications, where agents often rely on visual input. However, we believe that MENTOR is also applicable to state-based RL tasks. To support this, we conducted a fully state-based RL training experiment using Humanoid-Gym [9], where an agent must control a humanoid to accomplish a locomotion task. The comparison results, shown on the [rebuttal Section C](https://mentor-vrl.github.io/), indicate the effectiveness of our method.
>
> > Q3: Regarding the computational overhead of MoE layers:
>
> We thank the reviewer for pointing this out. We have discussed this in the original submission **Appendix E**: Time Efficiency of MENTOR, while MoE generally improves sample efficiency, it does introduce higher latency compared to standard MLPs due to its routing and expert selection. We acknowledge this tradeoff, but we want to highlight that the latency mostly comes from hardware optimization, not more computation overhead. Recent advances such as MoE-Infinity [10] offer promising directions to significantly reduce this overhead, potentially narrowing the gap to a negligible level.
>
> ----
> [5] Elsayed, M.,et.al. Streaming Deep Reinforcement Learning Finally Works. arXiv preprint arXiv:2410.14606.
>
> [6] Ji, T., Liang,et.al. Ace: Off-policy actor-critic with causality-aware entropy regularization. arXiv preprint arXiv:2402.14528.
>
> [7] Hafner, D.,et.al. Dream to control: Learning behaviors by latent imagination. arXiv preprint arXiv:1912.01603.
>
> [8] Laskin, M.,et.al. Curl: Contrastive unsupervised representations for reinforcement learning. In International conference on machine learning (pp. 5639-5650). PMLR.
>
> [9] Gu, X.,et.al. Humanoid-gym: Reinforcement learning for humanoid robot with zero-shot sim2real transfer. arXiv preprint arXiv:2404.05695.
>
> [10] Fu, Y.,et.al. MoE-CAP: Cost-Accuracy-Performance Benchmarking for Mixture-of-Experts Systems. arXiv preprint arXiv:2412.07067.

---

### Official Review · Reviewer_nuJ2 · 2025-03-18

**Overall Recommendation:** 4

**Summary:**

This paper introduces a MOE backbone approach to tackle multi-task visual RL, addressing the problem of conflicting gradients when training on completely opposite tasks (close vs open door). Experimental results show interestingly how the learned model switches between experts smoothly to tackle different sub-tasks of a problem, which further show evidence of some ability to learn across multiple tasks in a single model (with multiple experts). Neural network perturbation is also used to improve exploration of the RL process which mixes the weights with the current weights and some perturbation candidate weights. Sim and real world experiments are performed, with the real world training facilitated by some state estimation model to generate rewards.

**Claims And Evidence:**

Clear

**Essential References Not Discussed:**

N/A

**Experimental Designs Or Analyses:**

They are sound.

**Methods And Evaluation Criteria:**

They make sense and appropriate evaluations are made. Real experiments well appreciated.

**Other Comments Or Suggestions:**

- Figure 5 is missing error bars?

**Other Strengths And Weaknesses:**

Strengths:
- Great experimental results in simulation and real
- Clear ablation studies on the importance of MOE modelling to address the conflicting gradients problem for better multi-task RL
- Overall a well written paper

Weaknesses:
- Not a weakness of this specific paper but with real world RL there are way too many limitations to be practical sometimes that one should be aware of (real world RL in this case requires a dense/sparse reward function which requires a separate perception system usually to compute, and a auto-reset mechanism to be designed).
-

**Questions For Authors:**

- In the real world tasks are there any multi-task setups? Is one policy trained to do peg insertion, cable routing and the other tasks? I am wondering where MOE will be helpful in the real world experiments as they otherwise look like single-task.
- It seems MOE modeling is a big contribution of this paper, however it is unclear to me if the experiments in figure 6 really leverage MOE? Is it a single policy trained per task or one policy trained on all of those tasks? They seem quite a bit different from the motivating experiments with e.g. open/close door tasks in meta world. My best understanding is that some of the harder tasks have multiple sub-tasks which MOE is helping to learn on.
- The network perturbation process requires already trained expert policies / weights, how are those obtained? Specifically what is $\Phi_{\text{oriented}}$? Further reading suggests this is obtained online as training progresses but im not sure if I understood it correctly.
- Why specifically experts 9, 13, 14, and 15 are selected for visualization? Are they truly interpretable or is this possibly just some magical number/selection. What do the other experts look like?

Happy to raise score if above questions can be clarified.

**Relation To Broader Scientific Literature:**

They are related to past work on model-free visual RL, working towards more sample-efficient methods which can possibly enable faster real-world RL. Also related to some neural network research on network perturbation for avoiding local minima.

**Theoretical Claims:**

N/A

---

> ### Author Rebuttal · Authors · 2025-04-01
>
> > Q1: Regarding the weakness of real-world RL:
>
> We agree that real-world RL still faces several limitations. However, we have seen progress in this area. Recent work, such as Serl [1], has released a software suite that facilitates the rapid deployment of real-world learning paradigms. Moreover, recent advances in foundation models [2][3][4] can help simplify the design of perception systems for computing object-relevant rewards. In addition, we would like to emphasize that the main scope of our work is to propose a sample-efficient RL algorithm, which is not limited to real-world RL scenarios.
>
> > Q2: Figure 5 is missing error bars?
>
> Thanks for your comment, we will add it to the final camera-ready version.
>
> > Q3: In the real world tasks are there any multi-task setups?
>
> We trained three separate policies for three real-world tasks, each featuring specific challenges (details are provided in Section 4.2). In particular, the peg insertion task is designed to evaluate multi-task learning capability, as the policy must learn to insert different pegs into corresponding target holes using a single visual policy. We visualized the expert usage heatmap of MENTOR for this task in the original submission **Appendix G**, which shows that the MoE agent tends to utilize different experts for different plug shapes. The ablation study results shown in the original submission **Table 1** also demonstrate the effectiveness of the MoE structure compared to a standard MLP.
>
>
> > Q4: If the experiments in Figure 6 really leverage MOE? Can you show more multi-task experiments?
>
> Generally speaking, MENTOR is designed to improve the sample efficiency of visual RL, rather than focusing purely on multi-task scenarios. We found that MoE can promote learning efficiency by automatically leveraging distinct experts to decompose a hard task into several sub-tasks and learn them separately. Therefore, the experiments in Figure 6 are trained with a single policy per task. We really appreciate your valuable advice, so we conducted several multi-task learning experiments (detailed multi-task setup information is on [rebuttal Section A](https://mentor-vrl.github.io/)) to demonstrate MoE’s contribution in multi-task setups.
>
> > Q5: Explanation of Task-oriented Perturbation Mechanism:
>
> $\Phi_{\text{oriented}}$ is an approximate distribution over a set of high-performing agent weights $S_{\text{top}}$, which is constructed as training processes. Specifically, we maintain a fixed-size set $S_{\text{top}} = \{(\theta, R)\}$, which stores the weights $\theta$ and corresponding episode rewards $R$ of the top-performing agents seen so far. At each episode $t$, if the current agent’s reward $R_t$ exceeds the lowest reward in $S_{\text{top}}$, we replace the corresponding tuple with $(\theta_t, R_t)$. For task-oriented perturbation, we approximate $\Phi_{\text{oriented}}$ as a Gaussian $\mathcal{N}(\mu_\theta^{\text{top}}, \sigma_\theta^{\text{top}})$, where the mean and standard deviation are computed over the network weights in $S_{\text{top}}$. This dynamic update ensures that $\Phi_{\text{oriented}}$ remains focused on the most promising regions of the parameter space and generates more effective perturbation candidates $\phi$ compared with perturbation using random weights.
> A detailed description could be found in the original paper **Section 3.2**.
>
> > Q6: “Why specifically experts 9, 13, 14, and 15 are selected for visualization…”:
>
> The MoE agent used in this task (Meta-World Assembly) has 16 experts, with the top_k parameter set to 4. Therefore, in Figure 4, we mainly visualize the 4 most active experts during task execution, which happen to be experts 9, 13, 14, and 15. This selection may vary depending on the environment's random seed and the random initialization of expert weights. After training, the agent primarily uses these 4 experts, while the others exhibit low utilization during execution, so we did not include them in the visualization.
>
> -----
> [1] Luo, J.,et.al. Serl: A software suite for sample-efficient robotic reinforcement learning. In 2024 IEEE International Conference on Robotics and Automation (ICRA) (pp. 16961-16969). IEEE.
>
> [2] Liu, S.,et.al. Grounding dino: Marrying dino with grounded pre-training for open-set object detection. In European Conference on Computer Vision (pp. 38-55). Cham: Springer Nature Switzerland.
>
> [3] Wen, B.,et.al. Foundationpose: Unified 6d pose estimation and tracking of novel objects. In Proceedings of the IEEE/CVF Conference on Computer Vision and Pattern Recognition (pp. 17868-17879).
>
> [4] Wen, B.,et.al. FoundationStereo: Zero-Shot Stereo Matching. arXiv preprint arXiv:2501.09898.

---

> > ### Comment · Reviewer_nuJ2 · 2025-04-01
> >
> > Thanks for the detailed response and additional experiments.
> >
> > The majority of my concerns are addressed! I will raise my score to a 4.
> >
> > I still have some concern over interpretability figures like that in figure 4. If there really are only a few experts used, would it make sense to train with less experts? My interpretation would be if less experts does not do any better / does worse, than the interpretation of figure 4 might not make as much semantic sense as one might expect.

---

> > > ### Author Response · Authors · 2025-04-02
> > >
> > > Thanks for your reply! Generally speaking, we believe the change of top_k in Figure 4 have a significant impact on agent performance, while our method remains quite robust to the choice of the total number of experts in the MoE.
> > >
> > > From our empirical study, we found that a key factor significantly affecting agent performance is the number of experts allowed to be activated (i.e., top_k). To support this point, we refer you to [rebuttal Section B](https://mentor-vrl.github.io/): when the number of activated experts is insufficient to the task, increasing top_k can help improve performance (Curve 4_4 >> Curve 4_2 >> Curve 1_1). This indicates that the MoE multi-routing mechanism indeed plays an important role in agent performance, which is the main idea we aim to convey in Figure 4.
> > >
> > > Regarding the effect of the total number of experts in the MoE, we found the performance trend to be: Curve 8_4 > Curve 4_4 > Curve 32_4 (all of them >> Curve 4_2). This suggests that while the total number of experts can also influence performance, the agent's performance is generally robust to this parameter.

---

### Decision · Program_Chairs · 2025-05-01

**Decision:**

Accept (poster)

**Comment:**

This paper tackles multi-task reinforcement learning by utilizing a Mixture-of-Experts (MoE) backbone and introducing a task-oriented perturbation mechanism.

Strengths:

* Intuitive and reasonable idea to tackle hard problem
* Strong experimental results in both simulation and real-world settings
* Clear and well-structured writing

Weaknesses:
* Latency introduced by the MoE architecture
* Lack of theoretical analysis

Overall, I believe this paper makes a meaningful contribution and will be valuable to the community